# Hidden Activations Are Not Enough: A General Approach to Neural Network Predictions

## Abstract

We introduce a novel mathematical framework for analyzing neural networks using tools from quiver representation theory. This framework enables us to quantify the similarity between a new data sample and the training data, as perceived by the neural network. By leveraging the induced quiver representation of a data sample, we capture more information than traditional hidden layer outputs. This quiver representation abstracts away the complexity of the computations of the forward pass into a single matrix, allowing us to employ simple geometric and statistical arguments in a matrix space to study neural network predictions. Our mathematical results are architecture-agnostic and task-agnostic, making them broadly applicable. As proof of concept experiments, we apply our results for the MNIST and FashionMNIST datasets on the problem of detecting adversarial examples on different MLP architectures and several adversarial attack methods. Our experiments can be reproduced with our provided code.

## 1 Introduction

As neural networks increasingly dominate the machine learning landscape, a profound understanding of their behaviour and decision-making processes remains an elusive goal. Recent advances in quiver representation theory have provided a novel framework for analyzing neural networks, revealing a rich algebraic and geometric structure underlying these models. This paper builds upon the mathematical foundations established by Armenta & Jodoin (2021) and Armenta et al. (2022), which provide a profound mathematical bridge between neural networks and quiver representations.

We propose a theoretical framework for analyzing neural networks through the application of quiver representation theory. It has been shown by Armenta & Jodoin (2021) that a neural network produces a quiver representation for each input sample. Then, this quiver representation produces a matrix, as shown by Armenta et al. (2022). We show, by providing rigorous mathematical results, that these matrices can be used to study neural network data representations from a completely different perspective than what has been done in the literature, with vastly greater generality and algebraic consistency than only looking at hidden neuron activations.

For instance, we show that these matrices are invariant under isomorphisms of neural networks as defined by Armenta & Jodoin (2021), where an *isomorphism of neural networks* is defined as a transformation that is algebraically consistent with the neural network structure. It was also proved by Armenta & Jodoin (2021) that isomorphisms of neural networks preserve the network function, that is, isomorphic neural networks have exactly the same network function. They also proved that, although isomorphic neural networks have the same network function, their hidden activations can be completely different, rendering any analysis of the network in terms of the hidden activations algebraically inconsistent. This is the phenomenon exploited by the work of Dinh et al. (2017) to disprove the flat minima argument for the generalization capabilities of neural networks, which highlights the importance of taking into account this type of invariance when making theoretical claims about neural network behaviour.

As a proof of concept for our theoretical results, we present a simple algorithm for detecting adversarial examples using geometric and statistical arguments in matrix space. Roughly speaking, the matrix repre-

sentations generated by the neural network produce a hyper-ellipsoid per class (in classification or tokenized tasks) within this matrix space. We then use these hyper-ellipsoids to give a figure of merit for how close a new data sample is to the training distribution. Consequently, our detection method is architecture-agnostic and attack-method-agnostic, yielding a generalizable approach. Notably, this method proves to be more effective against certain attack methods, depending on the architecture.

While our adversarial example detection method based on this framework demonstrates promising results in identifying adversarial examples, its primary purpose is to illustrate the broader potential and applicability of the matrix statistics approach to neural networks. Through this work, we aim to encourage further exploration of the intersection between quiver representation theory, geometry and neural networks, ultimately advancing our understanding of these complex models and their role in machine learning.

In summary, our work makes the following key contributions:

1 We prove that the induced matrices of neural networks are invariant under isomorphisms of neural networks.

2 We prove that the distance between matrices in the infinity norm is greater or equal to the distance between the logits of the network in the maximum norm, independently of architecture, training or task.

3 We prove that the distance between matrices, considered as flattened vectors, in the 1-norm is greater or equal to the distance between the logits of the network in any $p$-norm, independently of architecture, training or task.

4 We provide a proof-of-concept adversarial example detection method based on these matrices, that only requires simple statistical and geometric arguments in matrix space.

5 We provide several experimental results of our detection method on different MLP architectures on the MNIST and FashionMNIST datasets with several adversarial attack methods, and we leverage our results transparent and reproducible via our provided code.

## 2 Previous Work

Since their appearance, neural networks have been studied in a per-layer fashion, see for instance the works by Hinton (2007); Goodfellow et al. (2016); Bronstein et al. (2021); Cammarata et al. (2020). However, a new perspective was introduced by Armenta & Jodoin (2021), where the neural network structure is broken down into pieces, consisting of fixed weights and activation functions that change during training but remain fixed during forward passes. This perspective allows the neural network to construct a quiver representation that contains all of the computations of the forward pass on a single input sample. This formalism is general and applies to any neural network, regardless of architecture, activation function, data, task, or learning algorithm. All of this with no approximation arguments, as done, for instance, in the spline theory of Balestriero & Baraniuk (2018), whose main results are restricted to piece-wise affine and convex activation functions and play a fundamental role in their proofs.

It is also shown in Section 6.1 of Armenta & Jodoin (2021) that neuron activations at a specific layer can be recovered from the induced quiver representation by composing with a forgetful functor, that is, by formally forgetting information in the language of category theory. These quiver representations are mapped to a moduli space and it was then proved by Armenta et al. (2022) that this moduli space can be embedded into the matrix space $\mathrm{Mat}_{\mathbb{R}}(k, d)$, where $d$ is the input space dimension and $k$ is the number of output neurons. As a consequence, a neural network maps every single input sample to a quiver representation and then to a matrix without losing any information, contrary to the hidden neuron outputs or latent spaces.

Furthermore, we prove that these matrices are invariant under isomorphisms of neural networks as defined by Armenta & Jodoin (2021). This is important because, for example, the positive scale invariance of ReLU is a particular case of isomorphism of neural networks, that was used by Dinh et al. (2017) to disprove flat minima arguments on generalization capabilities of neural networks made by Hochreiter & Schmidhuber (1997) and

Keskar et al. (2017), among others. Isomorphisms of neural networks, however, apply to any architecture independently of the activation function. The work of Dinh et al. (2017) suggests that understanding neural networks necessitates arguments that are, at the very least, invariant under positive scale invariance. Our results show that the induced matrices satisfy this property and are therefore suitable for investigating fundamental questions in the science of deep learning. This mathematical formalism and generality are precisely what is lacking in current machine learning research, and we aim to address this gap.

Neural networks are vulnerable to adversarial examples and out-of-distribution data, prompting the development of various detection methods. Robust training, model ensembles, and regular updates with performance monitoring are common approaches to enhance robustness, see for instance the work by Goodfellow et al. (2015); Shafahi et al. (2019); Deng & Mu (2024); Zhang et al. (2021b). The work of Hendrycks & Gimpel (2017) proposes several detection methods, including analyzing variance in PCA-whitened inputs and softmax distribution differences. Their approach involves examining the variance of coefficients in PCA-whitened inputs, leveraging the observation that adversarial images tend to have larger variance in later principal components compared to clean images. However, Carlini & Wagner (2017) found that this method can be bypassed if attackers are aware of it, as they can constrain the variance of later principal components during adversarial example generation. Hendrycks & Gimpel (2017) also suggested analyzing input reconstructions obtained by adding an auxiliary decoder to the classifier model. Metzen et al. (2017) introduced a binary detector network to differentiate between real and adversarial examples. However, our results suggest that methods based on network outputs, such as logits or softmax distributions, may not be as effective as those utilizing the matrices produced by the networks.

Recent research by Qi et al. (2023) has revealed a disturbing vulnerability in multimodal large language models (LLMs), where adversarial examples can be crafted to "jailbreak" these systems, circumventing their defences and raising alarming questions about the reliability, interpretability, and transparency of neural networks. This exposes a critical need for a rigorous, foundation-first approach to neural network design and one that prioritizes transparency, robustness, and mathematical rigour over black-box efficiency. Our work addresses this need, providing a foundation-first mathematical approach that sets a completely different view for studying and understanding neural networks.

## 3 Mathematical Background: Notation and Basic Results

For the reader's convenience, we gather all essential definitions and notation in this section, rendering the paper self-contained.

A *quiver* $Q = (Q_0, Q_1, s, t)$ is a quadruple consisting of an oriented graph $(Q_0, Q_1)$, with set of vertices $Q_0$ and set of oriented edges, or arrows, $Q_1$, and where $s, t : Q_1 \to Q_0$ are maps that associate to each arrow $\alpha \in Q_1$ its *source* $s(\alpha)$ and its *tail* $t(\alpha)$. A *representation* of $Q$ over a field $K$ is a couple $W = (W_q, W_\alpha)_{q \in Q_0, \alpha \in Q_1}$ where $W_q$ is a $K$-vector space and $M_\alpha : M_{s(\alpha)} \to M_{t(\alpha)}$ is a $K$-linear map. In this paper, we work with $K = \mathbb{R}$, the field of real numbers. Let $Q$ be a quiver and let $W$ and $V$ be two representations of $Q$. A collection of linear maps $\tau = (\tau_q)_{q \in Q_0} : W \to V$ is a *morphism* of representations if $\tau_{t(\alpha)} W_\alpha = V_\alpha \tau_{s(\alpha)}$ for all $\alpha \in Q_1$. We say that $\tau$ is an *isomorphism* if, for every $q \in Q_0$, $\tau_q$ is invertible. The representations $W$ and $V$ are *isomorphic* if there exists an isomorphism $\tau : W \to V$ and we write $W \cong V$.

In the context of neural networks, consider a quiver (graph) $Q$ representing a network with $d$ input neurons (source vertices) and $k$ output neurons (sink vertices). This quiver encodes the transformation of $d$-dimensional input data to a $k$-dimensional output space, which may correspond to $k$ classes in supervised classification or $k$ tokens in tokenized tasks.

When doing a feed-forward pass through a neural network with weights $W$ and activation function $f$ (that may, in principle, be different at each neuron), each individual weight in $W$ is used as multiplication by the weight, that is, if $W_\alpha$ is the weight of arrow $\alpha$, then the signal is passed from neuron $s(\alpha)$ to $t(\alpha)$ multiplied by $W_\alpha$. Since multiplication by a scalar is a linear map, this implies that the set of weights of the neural network defines a quiver representation. Following Armenta & Jodoin (2021), we define a *neural network* to be a pair $(W, f)$ where $W$ is a representation of $Q$ and $f = (f_q)_{q \in Q_0}$ a neuron-wise activation function. If $q$ is an input or output neuron, we set $f_q$ to be the identity function. Each neural network has its own induced

network function also known as its feed-forward function, which is a function that we denote by

$$\Psi(W, f) : \mathbb{R}^d \to \mathbb{R}^k.$$

Here, we consider the network function to output the logits of the network and not a distribution obtained after a softmax. The network function can be decomposed following Theorem 6.4 of Armenta & Jodoin (2021) and Theorem 3.3 of Armenta et al. (2022) in the following way. First, define the representation space of $Q$ as the set $\mathrm{Rep}(Q)$ formed by the quiver representations of $Q$ that have one-dimensional spaces assigned to each vertex (called thin quiver representations). Define now the *knowledge map*

$$\varphi(W, f) : \mathbb{R}^d \to \mathrm{Rep}(Q)$$

which depends on the neural network $(W, f)$ and associates to each input $x$ a quiver representation that we denote $\varphi(W, f)(x)$. This quiver representation has one-dimensional vector spaces in each vertex of the quiver and the linear maps associated to the arrows are given by

$$\Big(\varphi(W,f)(x)\Big)_\alpha = \begin{cases} W_\alpha x_{s(\alpha)} & \text{if } s(\alpha) \text{ is an input vertex,} \\ W_\alpha & \text{if } s(\alpha) \text{ is a bias vertex,} \\ W_\alpha \dfrac{a(W,f)_{s(\alpha)}(x)}{p(W,f)_{s(\alpha)}(x)} & \text{if } s(\alpha) \text{ is a hidden vertex,} \end{cases} \tag{1}$$

for every $\alpha \in Q_1$. Here, $a(W, f)_q(x)$ is the activation of neuron $q \in Q_0$ and $p(W, f)_q(x)$ is the corresponding pre-activation that we set to 1 when $q \in Q_0$ is an input neuron. We see that $(\varphi(W, f)(x))_\alpha$ is not well defined when $p(W, f)_{s(\alpha)}(x) = 0$. However, the set of all $x$ with this property is of measure zero, so we ignore them (see Remark 6.3 in (Armenta & Jodoin, 2021)). We label the input neurons of $Q$ from 1 to $d$, to make them correspond to the features of inputs in order. Also note that for bias vertices both activation and pre-activation are equal to 1. In this way, we can simply write $\Big(\varphi(W, f)(x)\Big)_\alpha = W_\alpha \dfrac{a(W,f)_{s(\alpha)}(x)}{p(W,f)_{s(\alpha)}(x)}$, for all $\alpha \in Q_1$.

Denote by $\Big(\varphi(W, f)(x), 1\Big)$ the neural network with fixed weights given by the quiver representation $\varphi(W, f)(x)$ and identity activation function at every neuron. Note that we do not train this neural network. Instead, as proved by Armenta & Jodoin (2021), for every input $x \in \mathbb{R}^d$, and letting $1_d = (1, ..., 1)^T \in \mathbb{R}^d$ we have the following result.

**Theorem 3.1.** $\Psi(W, f)(x) = \Psi\Big(\varphi(W, f)(x), 1\Big)(1_d)$.

This means that the network function factorizes through the space of representations $\mathrm{Rep}(Q)$ via the maps $\varphi(W, f) : \mathbb{R}^d \to \mathrm{Rep}(Q)$ and $\mathrm{ev}_{1_d} : \mathrm{Rep}(Q) \to \mathbb{R}^k$ where $\mathrm{ev}_{1_d}(V) = \Psi(V, 1)(1_d)$. In other words,

$$\Psi(W, f) = \mathrm{ev}_{1_d} \circ \varphi(W, f).$$

Note that the quiver representation $\varphi(W, f)(x)$ is a linear object, in which each layer is given by a matrix. This quiver representation is used to obtain the output of the network by evaluating $\Psi\Big(\varphi(W, f)(x), 1\Big)$, which is a linear map, and therefore it can be reduced from a composition of linear maps to a single matrix, given by the product of all these matrices as done in the work of Armenta et al. (2022). Observe that for more complicated architectures like DenseNets or transformers, the order of the product of these matrices has to be taken into account. We denote this map by $\pi : \mathrm{Rep}(Q) \to \mathrm{Mat}_{\mathbb{R}}(d, k)$, which is also known as tensor network contraction in many-body quantum systems (Cichocki et al., 2016). We denote the matrix induced by a neural network $(W, f)$ on an input sample $x$ by $\mathrm{M}(W, f)(x) = \pi \circ \varphi(W, f)(x)$. We therefore have that the network function decomposes as

$$\Psi(W, f) = \mathrm{ev}_1 \circ \pi \circ \varphi(W, f) = \mathrm{ev}_1 \circ \mathrm{M}(W, f)$$

where $\mathrm{ev}_1 : \mathrm{Mat}_{\mathbb{R}}(k, d) \to \mathbb{R}^k$ is the *evaluation map*, that sends a matrix $M$ to the product $M1_d$ where $1_d = (1, ..., 1)^T$.

We move now to the notion of isomorphisms of neural networks. Let $(W, f)$ and $(V, g)$ be two neural networks where $W$ and $V$ are representations of the same quiver $Q$. A *morphism* of neural networks $\tau : (W, f) \to (V, g)$ is a morphism of quiver representations $\tau : W \to V$ such that $\tau_q$ is the identity map when $q \in Q_0$ is a source or a sink vertex and $\tau_q f_q = g_q \tau_q$ for all $q \in Q_0$. We call a morphism $\tau : (W, f) \to (V, g)$ an *isomorphism* of neural networks if $\tau : W \to V$ is an isomorphism of representations. We say that two neural networks $(W, f)$ and $(V, g)$ are *isomorphic*, denoted $(W, f) \cong (V, g)$, if there exists an isomorphism $\tau : (W, f) \to (V, g)$.

The first property one can prove concerning isomorphic neural networks is Theorem 4.13 in the work of Armenta & Jodoin (2021), which states that the network function is invariant under isomorphisms, which means that isomorphic neural networks have exactly the same network function. This result implies that neural networks fit naturally into an algebraic context and are therefore suited to be studied by algebraic methods.

**Theorem 3.2.** *If $(W, f) \cong (V, g)$, then $\Psi(W, f)(x) = \Psi(V, g)(x)$ for every input $x \in \mathbb{R}^d$.*

# 4    Theoretical Results

In this section, we present our mathematical contributions, accompanied by rigorous proofs in Appendix B. Notably, our findings are universally applicable, as they do not rely on any assumptions regarding data distribution, network performance, or architectural design.

The first important property of all the previous constructions is their invariance under isomorphisms of neural networks. If we denote by $a(W, f)_q(x)$ the activation of neuron $q$ on the neural network $(W, f)$ when feeding the input $x$, and we take an isomorphism of neural networks $\tau : (W, f) \to (V, g)$, then, as a consequence of the proof of Theorem 4.13 by Armenta & Jodoin (2021) we have that

$$a(V, g)_q(x) = \tau_q a(W, f)_q(x).$$

This means that even though isomorphic neural networks have the same network function, their inner activations can be completely different. Since an isomorphism $\tau$ is determined by the choice of non-zero values $\tau_q$ per neuron $q$, there are infinitely many neural networks isomorphic to $(W, f)$ with this property. Therefore, invariance under isomorphism must be incorporated into any candidate of inner data representation of a neural network if one wants them to be algebraically consistent. Otherwise, any argument about the properties of a network using the outputs of hidden neurons is rendered inconsistent by applying any isomorphism to it.

Our findings uniformly exhibit this specific type of invariance, highlighting the vital importance of the upcoming theorem, given that all other invariants are contingent upon the knowledge map $\varphi(W, f)$.

**Theorem 4.1.** *If $(W, f) \cong (V, g)$, then for every $x$ the quiver representations $\varphi(W, f)(x)$ and $\varphi(V, g)(x)$ are isomorphic.*

The quiver representations $\varphi(W, f)(x)$ can be quite large to be saved in memory for several input samples $x$ as they are of the same size as the network. However, if one is interested in the prediction of the network, that is, the output of the network, then, as proved by Armenta et al. (2022), the quiver representation $\varphi(W, f)(x)$ can be mapped to a matrix via tensor network contraction (Cichocki et al., 2016), and the output of the network depends solely on this matrix. This mapping is an embedding, and therefore, no information is lost from the quiver representation $\varphi(W, f)(x)$. Our next result shows that the matrices induced by the quiver representations are invariant under isomorphisms, and therefore remain valid algebraic data representations of the neural network which are easier and less costly to manipulate than the whole quiver representations.

**Theorem 4.2.** *If $(W, f) \cong (V, g)$, then $\mathrm{M}(W, f)(x) = \mathrm{M}(V, g)(x)$ for all $x \in \mathbb{R}^d$.*

On a classification task (or on a tokenized task), the prediction is interpreted as the pre-assigned label (or token) of the output neuron with the highest value before the softmax activation. The softmax activation is a monotonic function which means the prediction is known from the logits and the softmax is used to transform the logits into probabilities. Therefore, multiplying the matrix $\mathrm{M}(W, f)(x)$ by the vector with only ones $1_d = (1, ..., 1) \in \mathbb{R}^d$ gives the logits of the network $(W, f)$, see Armenta et al. (2022). Then we can

partition the matrix space as follows. Define

$$\mathcal{M}_j = \left\{ M \in \mathrm{Mat}_{\mathbb{R}}(k,d) \; : \; \text{if we denote } M 1_d = \begin{pmatrix} a_1 \\ \vdots \\ a_k \end{pmatrix} \text{ then } a_j > a_i \text{ for all } j \neq i \right\}$$

We see from this that an input sample $x$ is classified as class $j$ by the network (independently of the probabilities obtained after applying the softmax to the logits) if and only if $\mathrm{M}(W,f)(x) \in \mathcal{M}_j$, and there is no clear decision if $\mathrm{M}(W,f)(x) \in \mathcal{M}_0$ where we have defined

$$\mathcal{M}_0 = \mathrm{Mat}_{\mathbb{R}}(k,d) - \bigcup_{j=1}^{k} \mathcal{M}_j.$$

In view of the next theorem, for the rest of the paper, we shall suppose that $\mathrm{M}(W,f)(x) \notin \mathcal{M}_0$ for every input $x$.

**Theorem 4.3.** *The set $\mathcal{M}_0$ is of measure zero.*

By the previous constructions, we obtain a decomposition of the matrix space

$$\mathrm{Mat}_{\mathbb{R}}(k,d) = \bigcup_{j=0}^{k} \mathcal{M}_j.$$

**Remark 4.1.** *The following theorem is the only mathematical result in this paper motivated by a specific task, either classification or tokenized task. The sole reason for this is that, in classification or tokenized tasks, a prediction is made based on choosing one output neuron, that is, the one with the highest value after the forward propagation. Note, however, that it is a result concerning only the matrix space and not neural networks.*

**Theorem 4.4.** *For each $j > 0$, the subset $\mathcal{M}_j$ is convex in the matrix space $\mathrm{Mat}_{\mathbb{R}}(k,d)$.*

In this case, convexity means that for matrices $A, B \in \mathcal{M}_j$ and for every scalar $\lambda \in [0,1]$ we have that $\lambda A + (1-\lambda)B \in \mathcal{M}_j$. Note that we do not make any reference to convexity of the parameter space of neural networks whatsoever.

Given this convexity property in the matrix space, we realize that if a neural network is well-trained on a classification task, that is, the accuracy both on train and test data is high (or the loss both on train and test is sufficiently low), then most of the training samples $x$ get mapped by $\mathrm{M}(W,f)$ to the convex subset of matrices $\mathcal{M}_j$ where $j$ is the correct label of $x$, for most $x$'s. Also, if we work only with correctly classified train data, they will all lie inside the corresponding subset $\mathcal{M}_j$.

**Remark 4.2.** *It is worth noting that the next theorem is highly general in that it is independent of the architecture, activation function, training algorithm used, achieved performance and nature of the data or the task. Even more, the network doesn't even have to be trained for it to hold. We put the proof here as it is simple enough and it illustrates how these matrices can be used to prove properties concerning norms and the network logits.*

**Theorem 4.5.** *Let $(W,f)$ be a neural network, $x, y \in \mathbb{R}^d$ be two inputs, and denote by $\mathbf{vec}(\mathrm{M}(W,f)(x))$ the vector form of $\mathrm{M}(W,f)(x)$. We have that*

1. $\| \mathrm{M}(W,f)(y) - \mathrm{M}(W,f)(x) \|_\infty \geq \| \Psi(W,f)(y) - \Psi(W,f)(x) \|_{\max}$, *and*

2. $\left\| \mathbf{vec}\Big( \mathrm{M}(W,f)(y) - \mathrm{M}(W,f)(x) \Big) \right\|_1 \geq \| \Psi(W,f)(y) - \Psi(W,f)(x) \|_p$, *for all vector $p$-norm, where $1 \leq p \leq \infty$.*

*Proof.* 1. By definition,

$$\| \operatorname{M}(W,f)(y) - \operatorname{M}(W,f)(x) \|_\infty = \sup_{z \in \mathbb{R}^d} \frac{\|\left( \operatorname{M}(W,f)(y) - \operatorname{M}(W,f)(x) \right) z\|_{\max}}{\|z\|_{\max}}$$

$$\geq \frac{\|\left( \operatorname{M}(W,f)(y) - \operatorname{M}(W,f)(x) \right) 1_d\|_{\max}}{\|1_d\|_{\max}}$$

$$= \| \operatorname{M}(W,f)(y)1_d - \operatorname{M}(W,f)(x)1_d \|_{\max}$$

$$= \| \Psi(W,f)(y) - \Psi(W,f)(x) \|_{\max}.$$

2. Using the triangle inequality and the fact that summing the elements of a row in $\operatorname{M}(W,f)(x)$ gives the corresponding logit,

$$\|\mathbf{vec}\left( \operatorname{M}(W,f)(y) - \operatorname{M}(W,f)(x) \right)\|_1 = \sum_{i_1=1}^{k} \sum_{i_2=1}^{d} | \operatorname{M}(W,f)(y)_{i_1,i_2} - \operatorname{M}(W,f)(x)_{i_1,i_2} |$$

$$\geq \sum_{i_1=1}^{k} | \sum_{i_2=1}^{d} \operatorname{M}(W,f)(y)_{i_1,i_2} - \operatorname{M}(W,f)(x)_{i_1,i_2} |$$

$$= \sum_{i_1=1}^{k} | \operatorname{M}(W,f)(y)_{i_1,:}1_d - \operatorname{M}(W,f)(x)_{i_1,:}1_d |$$

$$= \sum_{i_1=1}^{k} | \Psi(W,f)(y)_{i_1} - \Psi(W,f)(x)_{i_1} |$$

$$= \| \Psi(W,f)(y) - \Psi(W,f)(x) \|_1$$

$$\geq \| \Psi(W,f)(y) - \Psi(W,f)(x) \|_p.$$

$\square$

Using the previous result as a foundation, we devise a statistical figure of merit based on these matrices for a geometric picture of the training distribution, specifically class-dependent hyper-ellipsoids for a classification task. Considering $y = x + \epsilon$, where $\epsilon$ is adversarial noise, we try to detect adversarial examples by analyzing the matrix $\operatorname{M}(W,f)(y)$ and comparing it to the ones coming from correctly classified training samples $\operatorname{M}(W,f)(x)$.

## 5 Experiment Design

### 5.1 Trained Networks and Adversarial Attacks

We work with 8 multi-layered perceptrons with the hidden layer architectures listed in Table 1. We separately train these architectures on both datasets MNIST and FashionMNIST with different hyper-parameters achieving different performances that were not necessarily state-of-the-art, for a total of 17 trained networks. The performance and training hyperparameters of each one of these networks can be found in Appendix C. Here, we show results for 6 of these networks whose hyperparameters and performance are given in Table 2. For each one of the trained networks, we produced adversarial examples using the torchattacks library (Kim, 2020) with 21 different attack methods listed in Table 3. Note, however, that some of these adversarial attack methods do not appear in the tables of results for some of our bigger networks because they did not converge after days of computing. Particularly for architectures 7 and 8, which have 1.54 billion and 1.98 billion parameters, respectively.

Table 1: Each list in the middle represents the number of neurons per layer of different MLPs used in our experiments.

| Index | Hidden Layer Architecture | Number of Parameters |
|---|---|---|
| 1 | $5 \times [500]$ | 1.39 M |
| 2 | $8 \times [1000]$ | 7.79 M |
| 3 | $20 \times [1000]$ | 19.79 M |
| 4 | $[814, 351, 118, 467, 823, 191, 756, 628, 935, 270]$ | 2.9 M |
| 5 | $2 \times [10000]$ | 107.94 M |
| 6 | $5 \times [10000]$ | 407.94 M |
| 7 | $[675000, 1500, 1500, 1500, 1500]$ | 1.54 B |
| 8 | $[2500000]$ | 1.98 B |

Table 2: Training hyperparameters of our experiments shown in the main paper. Indexing corresponds to the experiments in our code (specified in the constants script). Here, Arch stands for the architecture index in Table 1; LR for learning rate; BS for batch size; LR Sch is an integer representing every how many epochs the learning rate is multiplied by 0.1.

| Experiment index | Arch | Optimizer | Dataset | LR | BS | Epochs | LR Sch | Accuracy train/test |
|---|---|---|---|---|---|---|---|---|
| 2 | 1 | Adam | Fashion | 1e-06 | 32 | 81 | 20 | 0.79/0.78 |
| 6 | 2 | Adam | MNIST | 0.001 | 128 | 6 | 3 | 0.98/0.97 |
| 10 | 5 | Momentum | Fashion | 0.01 | 32 | 11 | 5 | 0.96/0.9 |
| 12 | 6 | SGD | Fashion | 0.01 | 64 | 16 | - | 0.91/0.87 |
| 15 | 8 | Momentum | MNIST | 0.001 | 256 | 11 | - | 0.97/0.96 |
| 16 | 8 | Momentum | Fashion | 0.001 | 256 | 11 | 5 | 0.91/0.87 |

## 5.2 Detection Algorithm

Motivated by Theorem 4.5, which tells us that the distance between matrices is greater or equal to the output of the network, we work out a detection method for adversarial examples based on the induced matrices $M(W, f)(x)$ on training data $x$.

We shall write $(W, f)$ for a trained neural network and $\mathcal{D}$ for the dataset. Let $\mathcal{X} = \{x_i\}_{i=1}^N$ be a subset of $N = 10k$ uniformly drawn inputs of the dataset $\mathcal{D}$ and denote by $\mathcal{X}_j$ the subset of $\mathcal{X}$ containing the samples corresponding to class $j$. In our case, we took one thousand samples per $\mathcal{X}_j$. For each class $j$, we compute the *mean matrix* and the *standard deviation matrix*, defined as

$$M^j = \frac{1}{|\mathcal{X}_j|} \sum_{x \in \mathcal{X}_j} M(W,f)(x) \ \text{ and } \ S^j_{i_1,i_2} = \frac{1}{|\mathcal{X}_j| - 1} \left( \sum_{x \in \mathcal{X}_j} (M(W,f)(x)_{i_1,i_2} - M^j_{i_1,i_2})^2 \right), \quad (2)$$

respectively. Element-wise computation of means and standard deviations for each class produces hyper-ellipsoids in matrix space $\text{Mat}_\mathbb{R}(d, k) \cong \mathbb{R}^{dk}$ whose center is the mean and the axes are determined by the standard deviation in each coordinate. If, for every class $j$, we compute $M^j$ and $S^j$ with respect to correctly classified train data only, then it follows from Theorem 4.4 that the hyper-ellipsoid for class $j$ is contained in the subset $\mathcal{M}_j \subset \text{Mat}_\mathbb{R}(d, k)$ corresponding to its class.

Let $\mathcal{X}' = \{x_i'\}_{i=1}^{N'}$ be another subset of $N' = 10k$ uniformly drawn inputs of the dataset $\mathcal{D}$. For this case, we do not sample a specific amount of train samples for each class and rather take them at random from the whole training set. Let $\varepsilon, \varepsilon' > 0$. Suppose the neural network predicts that $x_i'$ is of class $j$. We denote by $n_i^\varepsilon(j)$ the number of entries of the matrix $M(W, f)(x_i')$ satisfying $(S^j)_{i_1, i_2} \leq \varepsilon$ and $M(W, f)(x_i')_{i_1, i_2} > \varepsilon$. To justify our interest in $n_i^\varepsilon(j)$, we found that there are several entries of the matrices of standard deviations

Table 3: Adversarial attack methods from torchattacks Kim (2020) used in our experiments.

| Abbreviation | Description |
|---|---|
| GN | Adds random noise from a Gaussian distribution to inputs. |
| FGSM | Perturbs inputs using gradients to maximize loss (Goodfellow et al., 2015). |
| RFGSM | Adds random noise before FGSM for variance (Tramèr et al., 2018). |
| PGD | Iteratively applies FGSM and projects onto epsilon-ball (Madry et al., 2018). |
| EOTPGD | Combines PGD with random transformations (Cohen et al., 2019). |
| FFGSM | Speeds up FGSM by scaling the gradient perturbation (Wong et al., 2020). |
| TPGD | PGD with theoretical guarantees (Zhang et al., 2019). |
| MIFGSM | Uses momentum to stabilize FGSM over multiple steps (Dong et al., 2018). |
| UPGD | Extends PGD for better performance in adversarial training. |
| DIFGSM | Varies inputs to improve transferability of attacks (Xie et al., 2019). |
| NIFGSM | Incorporates Nesterov momentum for faster convergence (Lin et al., 2020). |
| PGDRS | Combines PGD with randomized smoothing for defence (Cohen et al., 2019). |
| VMIFGSM | Adds variance to MIFGSM for robustness improvement (Wang & He, 2021). |
| VNIFGSM | Combines variance reduction with NIFGSM for better performance. |
| CW | Optimizes perturbations with a loss minimization (Carlini & Wagner, 2017). |
| PGDL2 | Uses L2 norm constraint in PGD for perturbation control (Madry et al., 2018). |
| PGDRSL2 | PGD Randomized Smoothing with an L2 norm constraint. |
| DeepFool | Minimizes perturbation to move inputs across decision boundaries (Moosavi-Dezfooli et al., 2016). |
| SparseFool | Generates sparse perturbations that alter few pixels (Modas et al., 2019). |
| OnePixel | Modifies one pixel at a time to create perturbations (Su et al., 2019). |
| Pixle | A pixel-based adversarial attack technique (Pomponi et al., 2022). |

Algorithm 1: Summary of the detection method described in Section 5.

---

**Algorithm 1** Detection method

**Require:** $\varepsilon, \varepsilon', t^\varepsilon, \mu^\varepsilon, \sigma^\varepsilon$, and an input $x_\ell$      ▷ *In our case, we did a grid search to obtain $\varepsilon, \varepsilon'$, and $t^\varepsilon$.*
1: **Compute:** $n_\ell^{\varepsilon'}(j)$
2: **if** $n_\ell^{\varepsilon'}(j) \geq \mu^\varepsilon - t^\varepsilon \sigma^\varepsilon$ **then**
3:      Trust the sample
4: **else**
5:      Reject the sample
6: **end if**

---

that are close to zero, which means that most of the samples have small variations along these coordinates. We remark that this may be because of the simplicity of the datasets and the networks and that more complicated datasets and networks may require different analyses.

We first compute the mean $\mu^\varepsilon$ and standard deviation $\sigma^\varepsilon$ of the set $\{n_i^\varepsilon(j)\}_{i=1}^{N'}$. Note that the notation $n_i^\varepsilon(j)$ means that the neural network predicts that $x_i$ is in class $j$, not that $x_i$ is labelled as class $j$. Then, given a threshold value $t^\varepsilon > 0$, we trust a new data sample $x_\ell$, classified as being of class $j$, if $n_\ell^{\varepsilon'}(j) \geq \mu^\varepsilon - t^\varepsilon \sigma^\varepsilon$, and reject it if not. In other words, we only trust the sample if enough entries of $\mathrm{M}(W, f)(x_\ell)$ are not close to zero. This is summarized in Algorithm 1.

# 6 Experimental Results

We ran a simple grid search on the parameters $\varepsilon, \varepsilon', t^\varepsilon$ for each individual network. We then choose the parameters $\varepsilon, \varepsilon', t^\varepsilon$ which give a higher difference between the percentage of good defences across all attack methods and wrong rejections on natural data. This worked for 15 of our experiments but for experiments 7 and 11 we couldn't find a high percentage of good defences that were considerably higher than the percentage of wrong rejections, so in their case, we chose parameters for which the percentage of good defences was high and the percentage of wrong rejections was strictly smaller. We did this, as sometimes one would prefer to have a network that rejects a lot of samples even when they are natural data to avoid adversarial examples passing unchecked, as in the case of LLM jailbreaks. Also note that different values of $\varepsilon, \varepsilon', t^\varepsilon$ work for different networks.

Here we show results for 6 different experiments and put the results for the rest in Appendix A. We can see that for each model, with the exception of experiments 7 and 11 as mentioned before, we can find parameters $\varepsilon, \varepsilon', t^\varepsilon$ that give high good defence probability and low wrong rejection probability. From all the results in the grid search, we take the top 3 results with the highest difference. We note that during the grid search, we found a lot of choices of the parameters in which the rejection level $\mu^\varepsilon - t^\varepsilon \sigma^\varepsilon$ was less than or equal to 0, therefore admitting every input as natural data. This shows there is no perfect choice for the parameters $\varepsilon, \varepsilon', t^\varepsilon$ and that this has to be done carefully. Also, there were choices of parameters for which both the good defence probability and the wrong rejection probabilities were both high and very close. This means that setting up the optimal values for $\varepsilon, \varepsilon', t^\varepsilon$ requires having access to some matrices of adversarial examples.

Nevertheless, we found that even in the best of cases, our detection algorithm is not perfect. Certain attacks like SparseFool, OnePixel and mainly Pixle seem to be good at fooling the detection method. Although in most of our experiments, we were not able to detect all of the adversarial examples coming from all the attacks, we do detect the majority of them while keeping a good rate of trusted natural data and a good accuracy on it. Also, we do not engineer specific detection methods for each attack method. Here, we show tables with detected adversarial examples and successful adversarial examples per attack method in Tables 4, 5, 6, 7, 8, 9, for each one of the 6 trained networks shown in Table 2.

# 7 Discussion

Our research addresses the need for mathematically rigorous approaches in deep learning, particularly in the context of providing a figure of merit to the training distribution as perceived by the neural network. Our proposed detection algorithm for adversarial examples demonstrates the potential of our matrix statistics derived from correctly classified training data to identify trustworthy samples. Besides the use of these mathematics to give this figure of merit, one could think of further applications that we expose in this section.

## 7.1 Out-of-Distribution Detection

Here, we propose a different detection method for the potential purpose of detecting samples coming from another dataset, i.e., out-of-distribution data. Suppose that $(W, f)$ is a neural network trained on a dataset $\mathcal{D}$. Assuming a classification task and that $(W, f)$ is trained and achieves high accuracy on $\mathcal{D}$, Theorem 4.4 tells us that almost every entry of $\mathrm{M}(W, f)(x)$ is in $\mathcal{M}_j$, where $j$ is the class of $x \in \mathcal{D}$. A natural test would be to look if, indeed, many entries are in $\mathcal{M}_j$.

Using again the matrices $M^j$ and $S^j$ of equation 2 and the set $\mathcal{X}' = \{x_i\}_{i=1}^{N'} \subseteq \mathcal{D}$ defined in Section 5, we could compute, for all $x_i \in \mathcal{X}'$, the number of entries $(i_1, i_2)$ of $\mathrm{M}(W, f)(x_i)$ satisfying $(M^j - \delta S^j)_{i_1, i_2} \leq \mathrm{M}(W, f)(x_i)_{i_1, i_2} \leq (M^j + \delta S^j)_{i_1, i_2}$ for a $\delta > 0$, which we denote by $n_i^\delta(j)$. Then, we compute the mean $\mu^\delta$

---

[1]We ran the detection algorithm on the test set. The elements of the test set were not altered. This number corresponds to how many samples of the test set were incorrectly classified as being adversarial examples.

[2]The accuracy under Trusted corresponds to accuracy on trusted data only, and the one at the right to accuracy on the whole test set.

Table 4: Experiment 2 - FashionMNIST - Architecture 1. $t^{\varepsilon} = 0.1$, $\varepsilon = 0.05$, and $\varepsilon' = 0.01$.

| Attack methods | Adversarial examples | | |
|---|---|---|---|
| | Detected | Successful | Total |
| GN | 4335 | 0 | 4335 |
| FGSM | 4973 | 0 | 4973 |
| RFGSM | 5052 | 0 | 5052 |
| PGD | 5051 | 0 | 5051 |
| EOTPGD | 5050 | 0 | 5050 |
| FFGSM | 4937 | 0 | 4937 |
| TPGD | 4885 | 0 | 4885 |
| MIFGSM | 5064 | 0 | 5064 |
| UPGD | 5064 | 0 | 5064 |
| DIFGSM | 4988 | 0 | 4988 |
| NIFGSM | 5320 | 0 | 5320 |
| PGDRS | 5038 | 0 | 5038 |
| VMIFGSM | 4998 | 0 | 4998 |
| VNIFGSM | 5062 | 0 | 5062 |
| CW | 5069 | 0 | 5069 |
| PGDL2 | 9004 | 0 | 9004 |
| PGDRSL2 | 4363 | 0 | 4363 |
| DeepFool | 6177 | 0 | 6177 |
| SparseFool | 6375 | 2122 | 8497 |
| OnePixel | 7852 | 2122 | 9974 |
| Pixle | 6 | 2339 | 2345 |
| | **Trusted** | **Wrongly rejected** | |
| **Test data** | 9801 | 199[1] | 10000 |
| **Accuracy**[2] | 0.78349 | - | 0.7887 |

Table 5: Experiment 6 - MNIST - Architecture 2. $t^{\varepsilon} = 0.75$, $\varepsilon = 0.1$, and $\varepsilon' = 0.01$.

| Attack methods | Adversarial examples | | |
|---|---|---|---|
| | Detected | Successful | Total |
| GN | 2878 | 200 | 3078 |
| FGSM | 3177 | 71 | 3248 |
| RFGSM | 3500 | 36 | 3536 |
| PGD | 3409 | 124 | 3533 |
| EOTPGD | 3094 | 439 | 3533 |
| FFGSM | 3428 | 46 | 3474 |
| TPGD | 3480 | 33 | 3513 |
| MIFGSM | 3432 | 84 | 3516 |
| UPGD | 3456 | 60 | 3516 |
| DIFGSM | 3410 | 39 | 3449 |
| NIFGSM | 3423 | 99 | 3522 |
| PGDRS | 3324 | 148 | 3472 |
| VMIFGSM | 3413 | 30 | 3443 |
| VNIFGSM | 3394 | 118 | 3512 |
| CW | 3455 | 51 | 3506 |
| PGDL2 | 9007 | 0 | 9007 |
| PGDRSL2 | 1756 | 1325 | 3081 |
| DeepFool | 4412 | 625 | 5037 |
| SparseFool | 6572 | 417 | 6989 |
| OnePixel | 8896 | 1104 | 10000 |
| Pixle | 78 | 301 | 379 |
| | **Trusted** | **Wrongly rejected** | |
| **Test data** | 7768 | 2232 | 10000 |
| **Accuracy** | 0.96910 | - | 0.9735 |

and the standard deviation $\sigma^{\delta}$ of the set $\{n_i^{\delta}(j)\}_{i=1}^{N'}$. Given an input $x_{\ell}$, potentially coming from a different dataset, and a threshold value $t^{\delta}$, we trust that $x_{\ell}$ is in $\mathcal{D}$ if $\mu^{\delta} - t^{\delta}\sigma^{\delta} \leq n_{\ell}^{\delta}(j) \leq \mu^{\delta} + t^{\delta}\sigma^{\delta}$.

For a stronger test, we could ask that $x_{\ell}$ passes the detection method described in Algorithm 1 before doing this other test.

## 7.2 Matrices of Subnetworks

Theorem 4.5 can be generalized to account for any subset of hidden layers. For instance, if we are interested in a subnetwork (or circuit in a transformer (Cammarata et al., 2020)) from layer $i$ to layer $j$, and we denote by $\Psi(W, f)_{i:j}$ the network function starting in layer $i$ up to layer $j$, then Theorem 4.5 holds for $\Psi(W, f)_{i:j}$. This could be applied to big networks in which computing the matrices coming from the whole network is too expensive, for example, on LLMs. On these huge models, one could calculate the matrices on layers at the beginning, middle and end either on the encoder or the decoder, to re-ask questions of mechanistic interpretability (Elhage et al., 2021a;b; Marks et al., 2024).

The fully connected layers of LLMs represent 2/3 of their parameters. Yet it is still an open problem to understand what happens inside these MLPs (Bricken et al., 2023; Elhage et al., 2022), for which our work provides tools. For instance, the middle layer of these MLPs is considerably bigger than their input and output and this makes it difficult to analyze. One could produce the matrices of these MLPs per input and compare the different matrices instead of looking at individual neurons in the middle layer. Our architecture 8 represents precisely this type of MLP for which experiments 15 and 16 in Tables 8 and 9 show good results of adversarial example detection on a few of the attack methods we used. The matrices can be computed

Table 6: Experiment 10 - FashionMNIST - Architecture 5. $t^\varepsilon = 10^{-5}$, $\varepsilon \approx 0.27826$, and $\varepsilon' \approx 0.02154$.

| Attack methods | Adversarial examples | | |
| --- | --- | --- | --- |
| | Detected | Successful | Total |
| GN | 4959 | 5 | 4964 |
| FGSM | 6309 | 3 | 6312 |
| RFGSM | 6891 | 2 | 6893 |
| PGD | 6024 | 866 | 6890 |
| EOTPGD | 6024 | 867 | 6891 |
| FFGSM | 5813 | 820 | 6633 |
| TPGD | 5932 | 805 | 6737 |
| MIFGSM | 5965 | 894 | 6859 |
| UPGD | 5965 | 894 | 6859 |
| DIFGSM | 5670 | 816 | 6486 |
| NIFGSM | 5979 | 986 | 6965 |
| PGDRS | 5937 | 879 | 6816 |
| VMIFGSM | 5913 | 862 | 6775 |
| VNIFGSM | 5965 | 894 | 6859 |
| CW | 5981 | 888 | 6869 |
| PGDL2 | 9004 | 0 | 9004 |
| PGDRSL2 | 0 | 5039 | 5039 |
| DeepFool | 8466 | 8 | 8474 |
| SparseFool | 8142 | 672 | 8814 |
| OnePixel | 7371 | 2629 | 10000 |
| Pixle | 398 | 816 | 1214 |
| | **Trusted** | **Wrongly rejected** | |
| **Test data** | 7172 | 2828 | 10000 |
| **Accuracy** | 0.91035 | - | 0.9049 |

Table 7: Experiment 12 - FashionMNIST - Architecture 6. $t^\varepsilon \approx 0.03480$, $\varepsilon \approx 0.22846$, and $\varepsilon' \approx 0.02154$.

| Attack methods | Adversarial examples | | |
| --- | --- | --- | --- |
| | Detected | Successful | Total |
| GN | 764 | 3 | 767 |
| FGSM | 974 | 8 | 982 |
| RFGSM | 1054 | 10 | 1064 |
| PGD | 1054 | 10 | 1064 |
| EOTPGD | 1054 | 10 | 1064 |
| FFGSM | 1027 | 8 | 1035 |
| TPGD | 1032 | 10 | 1042 |
| MIFGSM | 1050 | 10 | 1060 |
| UPGD | 1050 | 10 | 1060 |
| DIFGSM | 1040 | 8 | 1048 |
| NIFGSM | 1059 | 10 | 1069 |
| PGDRS | 1015 | 8 | 1023 |
| VMIFGSM | 1003 | 8 | 1011 |
| VNIFGSM | 1050 | 10 | 1060 |
| CW | 1048 | 10 | 1058 |
| PGDL2 | 9007 | 0 | 9007 |
| PGDRSL2 | 775 | 3 | 778 |
| DeepFool | 2702 | 14 | 2716 |
| SparseFool | 5745 | 183 | 5928 |
| OnePixel | 8277 | 1721 | 9998 |
| Pixle | 0 | 226 | 226 |
| | **Trusted** | **Wrongly rejected** | |
| **Test data** | 9921 | 79 | 10000 |
| **Accuracy** | 0.87841 | - | 0.8778 |

Table 8: Experiment 15 - MNIST - Architecture 8. $t^\varepsilon \approx 0.00530$, $\varepsilon \approx 0.22846$, and $\varepsilon' \approx 0.02154$.

| Attack methods | Adversarial examples | | |
| --- | --- | --- | --- |
| | Detected | Successful | Total |
| TPGD | 6334 | 0 | 6334 |
| MIFGSM | 6335 | 0 | 6335 |
| UPGD | 6335 | 0 | 6335 |
| DIFGSM | 6240 | 0 | 6240 |
| PGDRS | 6321 | 0 | 6321 |
| PGDRSL2 | 5837 | 0 | 5837 |
| | **Trusted** | **Wrongly rejected** | |
| **Test data** | 10000 | 0 | 10000 |
| **Accuracy** | 0.9619 | - | 0.9619 |

Table 9: Experiment 16 - FashionMNIST - Architecture 8. $t^\varepsilon = 1.25$, $\varepsilon = 0.8$, and $\varepsilon' \approx 0.27826$.

| Attack methods | Adversarial examples | | |
| --- | --- | --- | --- |
| | Detected | Successful | Total |
| GN | 6028 | 0 | 6028 |
| PGD | 7571 | 0 | 7571 |
| TPGD | 7306 | 0 | 7306 |
| PGDRSL2 | 6088 | 0 | 6088 |
| | **Trusted** | **Wrongly rejected** | |
| **Test data** | 9239 | 761 | 10000 |
| **Accuracy** | 0.88267 | | 0.8727 |

either on the forward pass of specific inputs and after neuron patching or ablation, and a similar figure of merit can be derived.

The lottery ticket hypothesis Frankle & Carbin (2019) states that some subnetworks can achieve the same performance as the original network or even better. The matrices for the lottery ticket can be compared to those of the whole network to identify meaningful differences between them.

### 7.3 Generalization

Generalization in deep learning is one of the biggest open problems (Zhang et al., 2017; 2021a; 2020). One of the ways of measuring generalization after training is to measure the difference between train and test performance (loss or accuracy). Using the matrices $M(W, f)(x)$ we can form the hyper-ellipsoids per class for each of the training set and the validation/test set. In this setting, generalization can be thought of as the hypervolume of the intersection of these hyper-ellipsoids. This brings a completely different perspective to measuring generalization than just looking at the performance metrics.

### 7.4 Comparing Architectures on the Same Task

Consider the case in which a task is fixed and we train different neural network architectures for it. Since the number of input and output neurons for all such networks are the same, the matrices they produce live in the same space since they will have the same dimensions. Even more, these matrices can be produced at different points during training to study the difference in the time evolution of the class-dependent hyper-ellipsoids for each architecture and their intersection with others. This provides a new way of comparing different architectures and their behaviour on the same task.

## 8 Conclusion

Our research addresses a critical gap in deep learning: the lack of mathematically rigorous results about how neural networks make predictions. As noted by Olah et al. (2020), interpretability is a field in which *there isn't consensus on what the objects of study are, what methods we should use to answer them, or how to evaluate research results.* This emphasizes the importance of a mathematically sound approach like ours, particularly invariance under isomorphisms, which renders neural networks and the matrices they induce algebraically consistent.

This work establishes a foundational framework for understanding neural network behaviour, providing rigorous mathematical insights into the geometric and algebraic structures underlying these complex models. Our theoretical contributions pave the way for a deeper understanding of neural networks, transcending specific architectures and applications.

By developing a mathematical theory of neural network behaviour, we can unlock new insights into the workings of these models and improve their reliability and security. Our research demonstrates the essential role of mathematical rigour in illuminating the inner workings of deep learning, providing a solid foundation for future advances in the field.

In conclusion, our work takes a crucial step toward bridging the gap between the empirical successes and theoretical understanding of deep learning. As we continue to explore the mathematical landscape of neural networks, we may uncover even more profound implications for the future of artificial intelligence.

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

Table 10: Experiment 0 - MNIST - Architecture 1. $t^\varepsilon = 1.25$, $\varepsilon = 0.05$, and $\varepsilon' = 0.05$.

Table 11: Experiment 1 - MNIST - Architecture 1. $t^\varepsilon = 0.1$, $\varepsilon = 1.25$, and $\varepsilon' = 0.01$.

| Attack methods | Adversarial examples | | |
|---|---|---|---|
| | Detected | Successful | Total |
| GN | 1208 | 3 | 1211 |
| FGSM | 1547 | 3 | 1550 |
| RFGSM | 1640 | 3 | 1643 |
| PGD | 1639 | 3 | 1642 |
| EOTPGD | 1640 | 3 | 1643 |
| FFGSM | 1583 | 3 | 1586 |
| TPGD | 1481 | 5 | 1486 |
| MIFGSM | 1641 | 3 | 1644 |
| UPGD | 1641 | 3 | 1644 |
| DIFGSM | 1585 | 3 | 1588 |
| NIFGSM | 1663 | 1 | 1664 |
| PGDRS | 1607 | 3 | 1610 |
| VMIFGSM | 1578 | 3 | 1581 |
| VNIFGSM | 1639 | 3 | 1642 |
| CW | 1637 | 3 | 1640 |
| PGDL2 | 9007 | 0 | 9007 |
| PGDRSL2 | 1196 | 3 | 1199 |
| DeepFool | 3486 | 5 | 3491 |
| SparseFool | 4872 | 847 | 5719 |
| OnePixel | 8741 | 1259 | 10000 |
| Pixle | 13 | 1042 | 1055 |
| | Trusted | Wrongly rejected | |
| Test data | 9065 | 935 | 10000 |
| Accuracy | 0.90723 | - | 0.9159 |

| Attack methods | Adversarial examples | | |
|---|---|---|---|
| | Detected | Successful | Total |
| GN | 1208 | 3 | 1211 |
| FGSM | 1547 | 3 | 1550 |
| RFGSM | 1640 | 3 | 1643 |
| PGD | 1639 | 3 | 1642 |
| EOTPGD | 1640 | 3 | 1643 |
| FFGSM | 1583 | 3 | 1586 |
| TPGD | 1481 | 5 | 1486 |
| MIFGSM | 1641 | 3 | 1644 |
| UPGD | 1641 | 3 | 1644 |
| DIFGSM | 1585 | 3 | 1588 |
| NIFGSM | 1663 | 1 | 1664 |
| PGDRS | 1607 | 3 | 1610 |
| VMIFGSM | 1578 | 3 | 1581 |
| VNIFGSM | 1639 | 3 | 1642 |
| CW | 1637 | 3 | 1640 |
| PGDL2 | 9007 | 0 | 9007 |
| PGDRSL2 | 1196 | 3 | 1199 |
| DeepFool | 3486 | 5 | 3491 |
| SparseFool | 4872 | 847 | 5719 |
| OnePixel | 8741 | 1259 | 10000 |
| Pixle | 13 | 1042 | 1055 |
| | Trusted | Wrongly rejected | |
| Test data | 9065 | 935 | 10000 |
| Accuracy | 0.98518 | - | 0.9835 |

Hongyang Zhang, Yaodong Yu, Jiantao Jiao, Eric Xing, Laurent El Ghaoui, and Michael Jordan. Theoretically principled trade-off between robustness and accuracy. In Kamalika Chaudhuri and Ruslan Salakhutdinov (eds.), *Proceedings of the 36th International Conference on Machine Learning*, volume 97 of *Proceedings of Machine Learning Research*, pp. 7472–7482. PMLR, 09–15 Jun 2019. URL https://proceedings.mlr.press/v97/zhang19p.html.

Xingwei Zhang, Xiaolong Zheng, and Wenji Mao. Adversarial perturbation defense on deep neural networks. *ACM Comput. Surv.*, 54(8), oct 2021b. ISSN 0360-0300. doi: 10.1145/3465397. URL https://doi.org/10.1145/3465397.

## Appendix A: Experimental Data

In this appendix, we provide extensive details on the results of our experiments, shown in Tables 10, 11, 12, 13, 14, 15,16, 17, 18, 19, and 20.

## Appendix B: Proofs

In this appendix, we suppose that $(W, f)$ and $(V, g)$ are neural networks with the same underlying quiver $Q$ and that data samples are in $\mathbb{R}^d$. We denote by $a(W, f)_q(x)$ the activation of neuron $q$ in the network $(W, f)$ when feed forwarding the input $x$, and by $p(W, f)_q(x)$ the corresponding pre-activation.

**Corollary of Theorem 4.13 by Armenta & Jodoin (2021).** *Let $\tau : (W, f) \to (V, g)$ be an isomorphism of neural networks and let $q \in Q_0$, then*

Table 12: Experiment 3 - FashionMNIST - Architecture 1. $t^\varepsilon = 0.1$, $\varepsilon = 0.8$, and $\varepsilon' = 0.05$.

| Attack methods | Adversarial examples | | |
|---|---|---|---|
| | Detected | Successful | Total |
| GN | 5439 | 769 | 6208 |
| FGSM | 5386 | 682 | 6068 |
| RFGSM | 7224 | 767 | 7991 |
| PGD | 7220 | 767 | 7987 |
| EOTPGD | 7217 | 770 | 7987 |
| FFGSM | 6938 | 800 | 7738 |
| TPGD | 7001 | 736 | 7737 |
| MIFGSM | 7132 | 790 | 7922 |
| UPGD | 7132 | 790 | 7922 |
| DIFGSM | 6420 | 822 | 7242 |
| NIFGSM | 6750 | 952 | 7702 |
| PGDRS | 6830 | 811 | 7641 |
| VMIFGSM | 6804 | 747 | 7551 |
| VNIFGSM | 7146 | 773 | 7919 |
| CW | 7069 | 746 | 7815 |
| PGDL2 | 9004 | 0 | 9004 |
| PGDRSL2 | 5475 | 797 | 6272 |
| DeepFool | 8021 | 886 | 8907 |
| SparseFool | 7388 | 1758 | 9146 |
| OnePixel | 7064 | 2922 | 9986 |
| Pixle | 439 | 942 | 1381 |
| | Trusted | Wrongly rejected | |
| Test data | 7946 | 2054 | 10000 |
| Accuracy | 0.91065 | - | 0.8984 |

Table 13: Experiment 4 - MNIST - Architecture 2. $t^\varepsilon = 0.1$, $\varepsilon = 0.01$, and $\varepsilon' = 0.01$.

| Attack methods | Adversarial examples | | |
|---|---|---|---|
| | Detected | Successful | Total |
| GN | 265 | 579 | 844 |
| FGSM | 856 | 181 | 1037 |
| RFGSM | 694 | 411 | 1105 |
| PGD | 921 | 184 | 1105 |
| EOTPGD | 921 | 184 | 1105 |
| FFGSM | 885 | 180 | 1065 |
| TPGD | 917 | 169 | 1086 |
| MIFGSM | 918 | 185 | 1103 |
| UPGD | 918 | 185 | 1103 |
| DIFGSM | 906 | 181 | 1087 |
| NIFGSM | 924 | 179 | 1103 |
| PGDRS | 856 | 190 | 1046 |
| VMIFGSM | 847 | 167 | 1014 |
| VNIFGSM | 918 | 185 | 1103 |
| CW | 911 | 183 | 1094 |
| PGDL2 | 9007 | 0 | 9007 |
| PGDRSL2 | 700 | 146 | 846 |
| DeepFool | 2145 | 380 | 2525 |
| SparseFool | 4500 | 1642 | 6142 |
| OnePixel | 8120 | 1872 | 9992 |
| Pixle | 69 | 188 | 257 |
| | Trusted | Wrongly rejected | |
| Test data | 7995 | 2005 | 10000 |
| Accuracy | 0.98086 | - | 0.981 |

Table 14: Experiment 5 - FashionMNIST - Architecture 2. $t^\varepsilon = 0.5$, $\varepsilon = 0.3$, and $\varepsilon' = 0.1$.

| Attack methods | Adversarial examples | | |
|---|---|---|---|
| | **Detected** | **Successful** | **Total** |
| GN | 3532 | 726 | 4258 |
| FGSM | 4509 | 931 | 5440 |
| RFGSM | 4743 | 1356 | 6099 |
| PGD | 4842 | 1254 | 6096 |
| EOTPGD | 5156 | 940 | 6096 |
| FFGSM | 5089 | 700 | 5789 |
| TPGD | 5084 | 747 | 5831 |
| MIFGSM | 5294 | 747 | 6041 |
| UPGD | 5306 | 735 | 6041 |
| DIFGSM | 5249 | 752 | 6001 |
| NIFGSM | 5143 | 901 | 6044 |
| PGDRS | 4934 | 835 | 5769 |
| VMIFGSM | 4839 | 854 | 5693 |
| VNIFGSM | 5282 | 760 | 6042 |
| CW | 5289 | 722 | 6011 |
| PGDL2 | 9004 | 0 | 9004 |
| PGDRSL2 | 3612 | 668 | 4280 |
| DeepFool | 6714 | 1075 | 7789 |
| SparseFool | 6916 | 1569 | 8485 |
| OnePixel | 2855 | 7141 | 9996 |
| Pixle | 0 | 1367 | 1367 |
| | **Trusted** | **Wrongly rejected** | |
| **Test data** | 5634 | 4366 | 10000 |
| **Accuracy** | 0.92829 | - | 0.8924 |

Table 15: Experiment 7 - FashionMNIST - Architecture 3. $t^\varepsilon \approx 0.00004$, $\varepsilon \approx 0.07743$, and $\varepsilon' \approx 0.02154$.

| Attack methods | Adversarial examples | | |
|---|---|---|---|
| | **Detected** | **Successful** | **Total** |
| GN | 4370 | 4591 | 8961 |
| FGSM | 5641 | 3350 | 8991 |
| RFGSM | 4366 | 5265 | 9631 |
| PGD | 5160 | 4473 | 9633 |
| EOTPGD | 5672 | 3961 | 9633 |
| FFGSM | 5307 | 4249 | 9556 |
| TPGD | 5214 | 4314 | 9528 |
| MIFGSM | 5676 | 3946 | 9622 |
| UPGD | 5004 | 4618 | 9622 |
| DIFGSM | 5668 | 3907 | 9575 |
| NIFGSM | 4630 | 4999 | 9629 |
| PGDRS | 4522 | 4909 | 9431 |
| VMIFGSM | 4264 | 5159 | 9423 |
| VNIFGSM | 4674 | 4944 | 9618 |
| CW | 5611 | 3882 | 9493 |
| PGDL2 | 9004 | 0 | 9004 |
| PGDRSL2 | 5276 | 3720 | 8996 |
| DeepFool | 5955 | 3714 | 9669 |
| SparseFool | 4892 | 4862 | 9754 |
| OnePixel | 5031 | 4968 | 9999 |
| Pixle | 761 | 1268 | 2029 |
| | **Trusted** | **Wrongly rejected** | |
| **Test data** | 8917 | 1083 | 10000 |
| **Accuracy** | 0.84356 | - | 0.8304 |

Table 16: Experiment 8 - MNIST - Architecture 3. $t^\varepsilon = 0.75$, $\varepsilon = 0.01$, and $\varepsilon' = 0.01$.

| Attack methods | Adversarial examples | | |
| --- | --- | --- | --- |
| | Detected | Successful | Total |
| GN | 2450 | 80 | 2530 |
| FGSM | 2644 | 98 | 2742 |
| RFGSM | 2849 | 120 | 2969 |
| PGD | 2851 | 118 | 2969 |
| EOTPGD | 2851 | 118 | 2969 |
| FFGSM | 2813 | 110 | 2923 |
| TPGD | 2813 | 129 | 2942 |
| MIFGSM | 2850 | 112 | 2962 |
| UPGD | 2850 | 112 | 2962 |
| DIFGSM | 2840 | 108 | 2948 |
| NIFGSM | 2856 | 109 | 2965 |
| PGDRS | 2775 | 116 | 2891 |
| VMIFGSM | 2742 | 109 | 2851 |
| VNIFGSM | 2849 | 113 | 2962 |
| CW | 2831 | 117 | 2948 |
| PGDL2 | 9007 | 0 | 9007 |
| PGDRSL2 | 2467 | 88 | 2555 |
| DeepFool | 4210 | 211 | 4421 |
| SparseFool | 6875 | 663 | 7538 |
| OnePixel | 7220 | 2776 | 9996 |
| Pixle | 8 | 317 | 325 |
| | Trusted | Wrongly rejected | |
| Test data | 9106 | 894 | 10000 |
| Accuracy | 0.96749 | - | 0.9689 |

Table 17: Experiment 9 - MNIST - Architecture 5. $t^\varepsilon = 0.25$, $\varepsilon = 0.3$, and $\varepsilon' = 0.01$.

| Attack methods | Adversarial examples | | |
| --- | --- | --- | --- |
| | Detected | Successful | Total |
| GN | 1005 | 1 | 1006 |
| FGSM | 1292 | 1 | 1293 |
| RFGSM | 1364 | 1 | 1365 |
| PGD | 1365 | 1 | 1366 |
| EOTPGD | 1363 | 1 | 1364 |
| FFGSM | 1322 | 1 | 1323 |
| TPGD | 1314 | 1 | 1315 |
| MIFGSM | 1353 | 1 | 1354 |
| UPGD | 1353 | 1 | 1354 |
| DIFGSM | 1349 | 1 | 1350 |
| NIFGSM | 1389 | 2 | 1391 |
| PGDRS | 1361 | 1 | 1362 |
| VMIFGSM | 1346 | 1 | 1347 |
| VNIFGSM | 1353 | 1 | 1354 |
| CW | 1363 | 1 | 1364 |
| PGDL2 | 9007 | 0 | 9007 |
| PGDRSL2 | 993 | 1 | 994 |
| DeepFool | 2911 | 7 | 2918 |
| SparseFool | 5369 | 356 | 5725 |
| OnePixel | 9331 | 645 | 9976 |
| Pixle | 10 | 440 | 450 |
| | Trusted | Wrongly rejected | |
| Test data | 9930 | 70 | 10000 |
| Accuracy | 0.96536 | - | 0.9668 |

Table 18: Experiment 11 -MNIST - Architecture 6. $t^\varepsilon = 0.00002$, $\varepsilon = 0.01172$, and $\varepsilon' = 0.01172$.

| Attack methods | Adversarial examples | | |
| --- | --- | --- | --- |
| | Detected | Successful | Total |
| GN | 281 | 2612 | 2893 |
| FGSM | 3005 | 292 | 3297 |
| RFGSM | 3137 | 264 | 3401 |
| PGD | 3137 | 265 | 3402 |
| EOTPGD | 3137 | 264 | 3401 |
| FFGSM | 3073 | 261 | 3334 |
| TPGD | 3131 | 246 | 3377 |
| MIFGSM | 3128 | 271 | 3399 |
| UPGD | 3128 | 271 | 3399 |
| DIFGSM | 3039 | 261 | 3300 |
| NIFGSM | 3178 | 283 | 3461 |
| PGDRS | 3124 | 271 | 3395 |
| VMIFGSM | 3081 | 270 | 3351 |
| VNIFGSM | 3124 | 273 | 3397 |
| CW | 3126 | 271 | 3397 |
| PGDL2 | 9007 | 0 | 9007 |
| PGDRSL2 | 277 | 2574 | 2851 |
| DeepFool | 5123 | 332 | 5455 |
| Pixle | 247 | 428 | 675 |
| | Trusted | Wrongly rejected | |
| Test data | 7712 | 2288 | 10000 |
| Accuracy | 0.95236 | - | 0.9464 |

- $a(V, g)_q(x) = \tau_q a(W, f)_q(x)$,

- $p(V, g)_q(x) = \tau_q p(W, f)_q(x)$.

**Theorem 4.1.** *If $(W, f) \cong (V, g)$, then for every $x \in \mathbb{R}^d$ the quiver representations $\varphi(W, f)(x)$ and $\varphi(V, g)(x)$ are isomorphic.*

*Proof.* Let $\tau : (W, f) \to (V, g)$ be an isomorphism of neural networks. We have that

$$
\begin{aligned}
\left(\varphi(V, g)(x)\right)_\alpha &= V_\alpha \frac{a(V, g)_{s(\alpha)}(x)}{p(V, g)_{s(\alpha)}(x)} \\
&= V_\alpha \frac{\tau_{s(\alpha)} a(W, f)_{s(\alpha)}(x)}{\tau_{s(\alpha)} p(W, f)_{s(\alpha)}(x)} \\
&= V_\alpha \frac{a(W, f)_{s(\alpha)}(x)}{p(W, f)_{s(\alpha)}(x)} \\
&= \frac{V_\alpha}{W_\alpha} \left(\varphi(W, f)(x)\right)_\alpha .
\end{aligned}
$$

Since $\tau_{t(\alpha)} W_\alpha = V_\alpha \tau_{s(\alpha)}$, we get that

$$
\begin{aligned}
\left(\varphi(V, g)(x)\right)_\alpha \tau_{s(\alpha)} &= \tau_{s(\alpha)} \frac{V_\alpha}{W_\alpha} \left(\varphi(W, f)(x)\right)_\alpha \\
&= \tau_{t(\alpha)} \frac{W_\alpha}{W_\alpha} \left(\varphi(W, f)(x)\right)_\alpha \\
&= \tau_{t(\alpha)} \left(\varphi(W, f)(x)\right)_\alpha .
\end{aligned}
$$

This means that $\tau : \varphi(W, f)(x) \to \varphi(V, g)(x)$ defines the required isomorphism. $\qquad \square$

Table 19: Experiment 13 - MNIST - Architecture 7. $t^\varepsilon \approx 0.00046$, $\varepsilon \approx 0.27826$, and $\varepsilon' = 1.0$.

| Attack | Adversarial examples | | |
|---|---|---|---|
| methods | Detected | Successful | Total |
| GN | 3813 | 0 | 3813 |
| FGSM | 4102 | 0 | 4102 |
| RFGSM | 4226 | 0 | 4226 |
| PGD | 4226 | 0 | 4226 |
| EOTPGD | 4226 | 0 | 4226 |
| FFGSM | 4180 | 0 | 4180 |
| TPGD | 4219 | 0 | 4219 |
| MIFGSM | 4211 | 0 | 4211 |
| UPGD | 4211 | 0 | 4211 |
| DIFGSM | 0 | 4220 | 4220 |
| NIFGSM | 4262 | 0 | 4262 |
| PGDRS | 0 | 4204 | 4204 |
| VMIFGSM | 3783 | 405 | 4188 |
| VNIFGSM | 0 | 4211 | 4211 |
| CW | 4102 | 108 | 4210 |
| PGDL2 | 9007 | 0 | 9007 |
| PGDRSL2 | 0 | 3805 | 3805 |
| DeepFool | 5579 | 0 | 5579 |
| Pixle | 393 | 0 | 393 |
| | **Trusted** | **Wrongly** | |
| | | **rejected** | |
| **Test data** | 7480 | 2520 | 10000 |
| **Accuracy** | 0.97099 | - | 0.9705 |

Table 20: Experiment 14 - FashionMNIST - Architecture 7. $t^\varepsilon \approx 0.00046$, $\varepsilon \approx 0.27826$, and $\varepsilon' = 1.0$.

| Attack | Adversarial examples | | |
|---|---|---|---|
| methods | Detected | Successful | Total |
| GN | 5122 | 0 | 5122 |
| FGSM | 6116 | 160 | 6276 |
| RFGSM | 6246 | 336 | 6582 |
| PGD | 4116 | 2470 | 6586 |
| EOTPGD | 6415 | 168 | 6583 |
| MIFGSM | 3180 | 3384 | 6564 |
| UPGD | 6564 | 0 | 6564 |
| DIFGSM | 0 | 6330 | 6330 |
| NIFGSM | 6568 | 172 | 6740 |
| PGDRS | 2296 | 4232 | 6528 |
| VMIFGSM | 5482 | 984 | 6466 |
| VNIFGSM | 6229 | 336 | 6565 |
| CW | 6231 | 328 | 6559 |
| PGDL2 | 9004 | 0 | 9004 |
| PGDRSL2 | 0 | 5104 | 5104 |
| DeepFool | 847 | 7082 | 7929 |
| Pixle | 1759 | 0 | 1759 |
| | **Trusted** | **Wrongly** | |
| | | **rejected** | |
| **Test data** | 6220 | 3780 | 10000 |
| **Accuracy** | 0.85965 | - | 0.8629 |

**Theorem 4.2.** *If $(W, f) \cong (V, g)$, then $\mathrm{M}(W, f)(x) = \mathrm{M}(V, g)(x)$ for all $x \in \mathbb{R}^d$.*

*Proof.* If the quiver representations $\varphi(W, f)(x)$ and $\varphi(V, g)(x)$ are isomorphic then, by definition, they define the same point in the moduli space of the neural network, and by Lemma 7.4 of Armenta et al. (2022) we conclude that $\mathrm{M}(W, f)(x) = \mathrm{M}(V, g)(x)$ for all $x$. $\qquad\square$

**Theorem 4.3.** *The set $\mathcal{M}_0$ is of measure zero.*

*Proof.* The set $\mathcal{M}_0$ is composed of all matrices $M = (m_{i_1, i_2}) \in \mathrm{Mat}_{\mathbb{R}}(k, d) \cong \mathbb{R}^{dk}$ such that at least two rows of $M$ sum to the same number. Therefore, $\mathcal{M}_0$ is the union of $\binom{dk}{2}$ linear subspaces of dimension $dk - 1$, given by an equation of the form $\sum_{p=1}^{d} m_{i_1, p} - \sum_{p=1}^{d} m_{i'_1, p} = 0$ for $i_1 \neq i'_1$. It is a well-known fact that an hyperplane is of (Lebesgue) measure zero and that a finite union of sets of measure zero is of measure zero. The result follows. $\qquad\square$

**Theorem 4.4.** *$\mathcal{M}_j$ is a convex subset of $\mathrm{Mat}_{\mathbb{R}}(k, d)$.*

*Proof.* Let $A \in \mathrm{Mat}_{\mathbb{R}}(k, d)$ be a matrix, and let $\mathbf{v}(A) \in \mathbb{R}^k$ be the vector obtained by summing the elements of each row of $A$, that is, $\mathbf{v}(A) = A 1_d$. Specifically, the $i$-th coordinate of $\mathbf{v}(A)$ is given by:

$$\mathbf{v}(A)_i = \sum_{p=1}^{d} A_{i,p}.$$

To show that $\mathcal{M}_j$ is convex, we need to prove that for any two matrices $A, B \in \mathcal{M}_j$ and any $\lambda \in [0, 1]$, the matrix $C = \lambda A + (1 - \lambda) B$ also belongs to $\mathcal{M}_j$. Given $A, B \in \mathcal{M}_j$, we denote the coordinates of $\mathbf{v}(A)$ and $\mathbf{v}(B)$, respectively, by:

$$\mathbf{v}(A) = [v_1(A), v_2(A), \dots, v_k(A)]$$
$$\mathbf{v}(B) = [v_1(B), v_2(B), \dots, v_k(B)]$$

By the definition of $\mathcal{M}_j$, we know that:

$$v_j(A) = \max(v_1(A), v_2(A), \dots, v_k(A))$$

$$v_j(B) = \max(v_1(B), v_2(B), \dots, v_k(B))$$

Consider the matrix $C = \lambda A + (1 - \lambda) B$. The sum of the $i$-th row of $C$ is:

$$v_i(C) = \sum_{p=1}^{d} C_{i,p} = \sum_{p=1}^{d} (\lambda A_{i,p} + (1 - \lambda) B_{i,p}) = \lambda \sum_{p=1}^{d} A_{i,p} + (1 - \lambda) \sum_{p=1}^{d} B_{i,p}$$

Therefore:

$$v_i(C) = \lambda v_i(A) + (1 - \lambda) v_i(B)$$

We need to show that the maximum coordinate of $\mathbf{v}(C)$ is also at index $j$. Since $v_j(A)$ and $v_j(B)$ are the maximum coordinates of $\mathbf{v}(A)$ and $\mathbf{v}(B)$ respectively, we have:

$$v_j(A) \geq v_i(A) \text{ for all } i, \text{ and}$$

$$v_j(B) \geq v_i(B) \text{ for all } i.$$

Thus,

$$\lambda v_j(A) + (1 - \lambda) v_j(B) \geq \lambda v_i(A) + (1 - \lambda) v_i(B) \text{ for all } i$$

This implies that $v_j(C) \geq v_i(C)$ for all $i$, meaning that the maximum coordinate of $\mathbf{v}(C)$ is at index $j$. Hence, the subset $\mathcal{M}_j$ is convex. $\qquad\square$

## Appendix C: Training regime

This appendix provides the specific hyperparameters used to train our networks and the performance they achieve. The hyperparameters are given in Table 22. The training curves in Table 21 and Figures 1, 2, 3 and 4.

Table 21: These experiments were trained for one epoch only so we do not show training curves.

| Experiment index | Train accuracy / Test accuracy | Train loss / Test loss |
|---|---|---|
| 0 | 0.9163 / 0.9159 | 2.1185 / 2.1067 |
| 9 | 0.9742 / 0.9668 | 2.755 / 3.2392 |

Table 22: Training hyperparameters of our experiments. Here, Exp stands for the experiment index; Arch for the architecture index in table 1; LR for learning rate; BS for batch size; Ep for epochs; LR Sch is an integer representing every how many number of epochs the learning rate is reduced by multiplying it by 0.1; WD for weight decay; Dpt for dropout probabilities

| Experiment | Architecture | Optimizer | Dataset | LR | BS | Epochs | LR Sch | WD | Dpt |
|---|---|---|---|---|---|---|---|---|---|
| 0 | 1 | SGD | MNIST | 0.01 | 8 | 1 | - | - | - |
| 1 | 1 | Momentum | MNIST | 0.01 | 32 | 11 | 5 | - | - |
| 2 | 1 | Adam | Fashion | 1e-06 | 32 | 81 | 20 | - | - |
| 3 | 1 | SGD | Fashion | 0.1 | 16 | 51 | 20 | - | - |
| 4 | 2 | Momentum | MNIST | 0.01 | 32 | 7 | 5 | - | - |
| 5 | 2 | Momentum | Fashion | 0.01 | 32 | 11 | 5 | - | - |
| 6 | 2 | Adam | MNIST | 0.001 | 128 | 6 | 3 | - | - |
| 7 | 3 | Adam | Fashion | 0.0001 | 16 | 11 | 5 | - | - |
| 8 | 4 | Adam | MNIST | 0.001 | 128 | 6 | - | - | - |
| 9 | 5 | Momentum | MNIST | 0.01 | 32 | 1 | - | - | - |
| 10 | 5 | Momentum | Fashion | 0.01 | 32 | 11 | 5 | - | - |
| 11 | 6 | SGD | MNIST | 0.01 | 64 | 6 | - | - | - |
| 12 | 6 | SGD | Fashion | 0.01 | 64 | 16 | - | - | - |
| 13 | 7 | Momentum | MNIST | 0.001 | 128 | 11 | - | 1e-05 | 0.5 |
| 14 | 7 | Momentum | Fashion | 0.001 | 128 | 11 | - | 1e-05 | 0.5 |
| 15 | 8 | Momentum | MNIST | 0.001 | 256 | 11 | - | - | - |
| 16 | 8 | Momentum | Fashion | 0.001 | 256 | 11 | 5 | - | - |

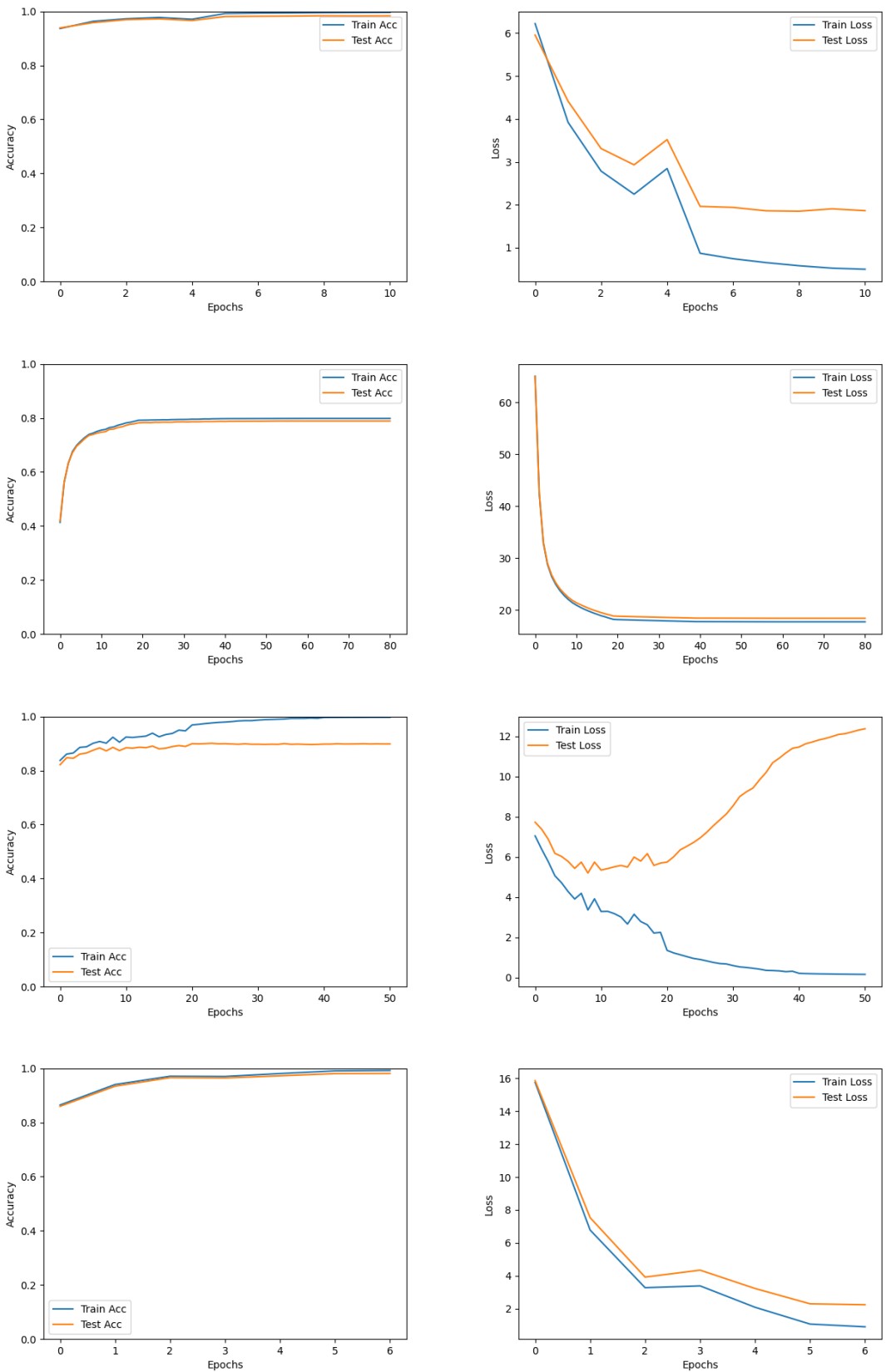

Figure 1: Training curves of experiments 1, 2, 3 and 4 from top to bottom.

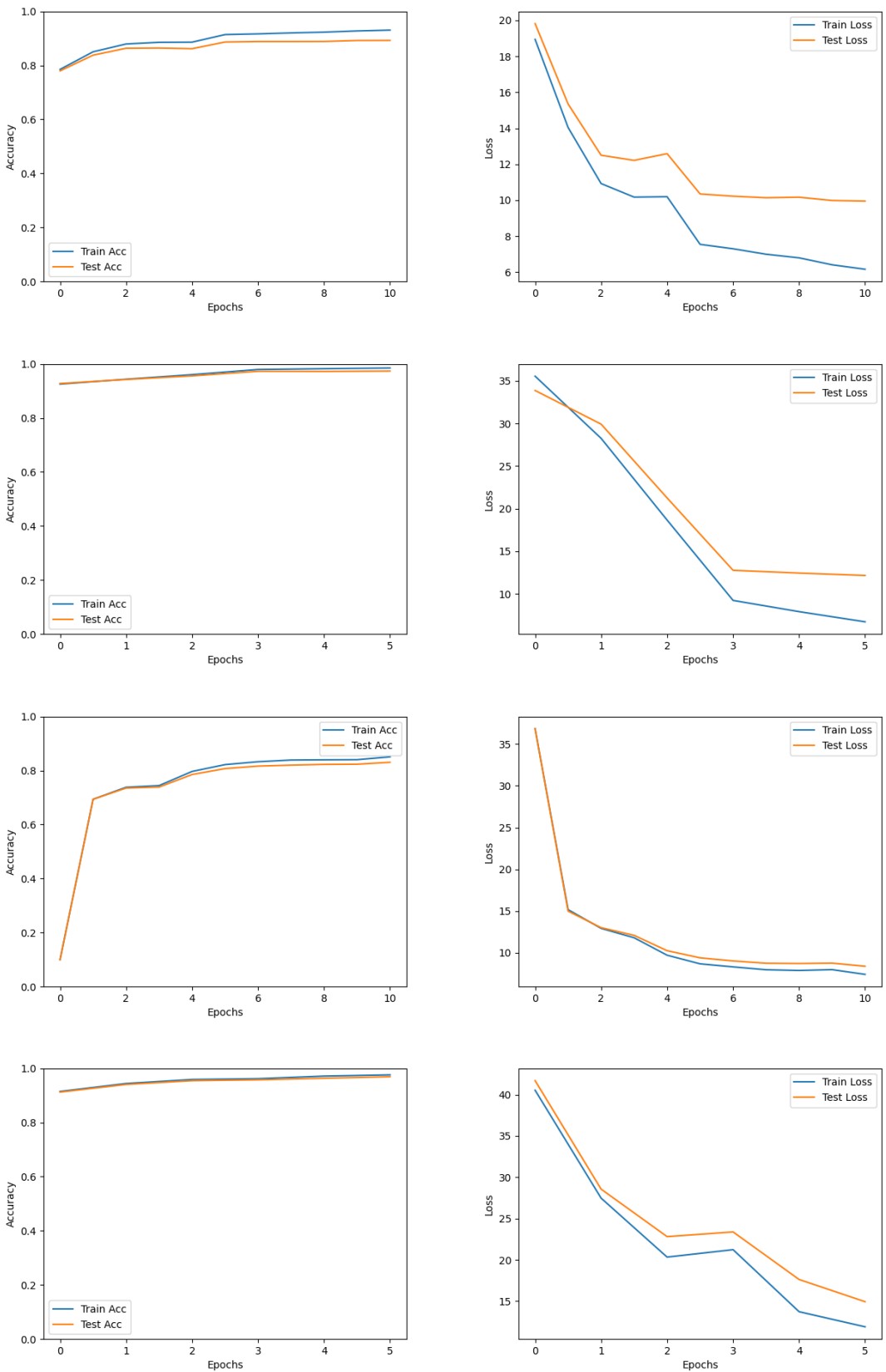

Figure 2: Training curves of experiments 5, 6, 7 and 8 from top to bottom.

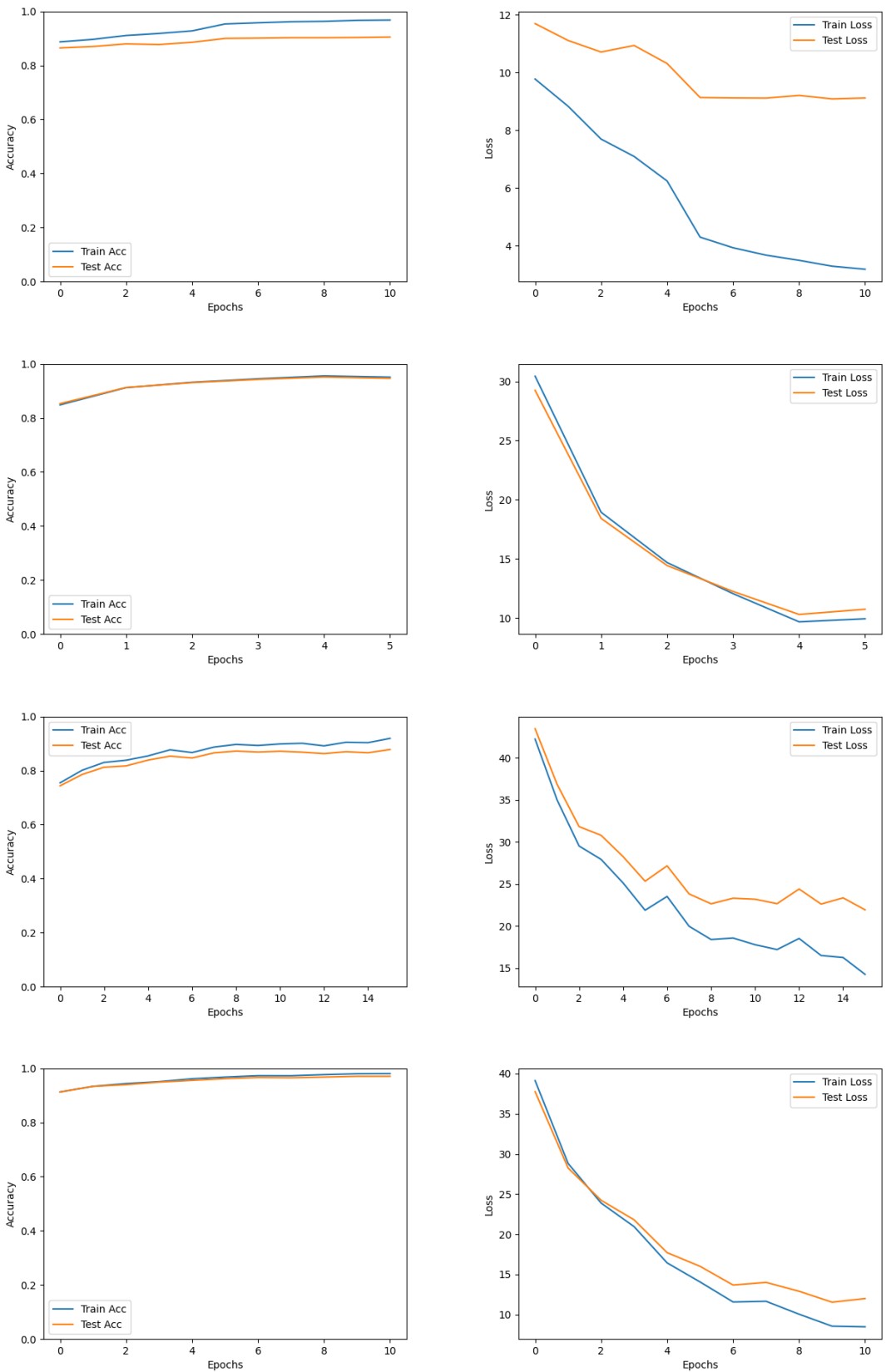

Figure 3: Training curves of experiments 10, 11, 12 and 13 from top to bottom.

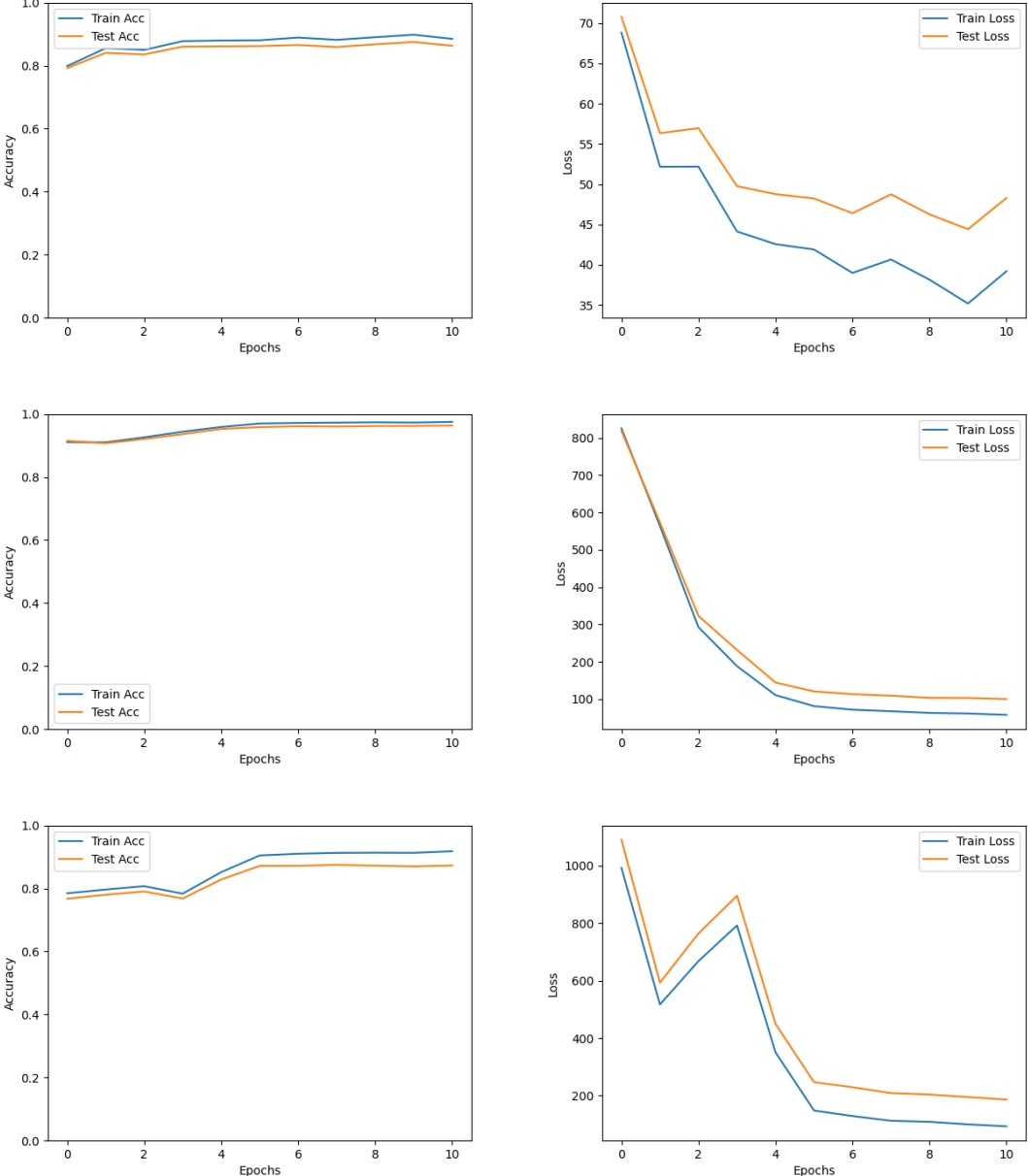

Figure 4: Training curves of experiments 14, 15 and 16 from top to bottom.

