# OpenReview forum: "Hidden Activations Are Not Enough: A General Approach to Neural Network Predictions"
_TMLR — Rejected by TMLR_

### Review · Reviewer_Vj72 · 2024-09-24

**Summary Of Contributions:**

This paper introduces an approach to anomaly detection that, as far as I know, is very novel. It provides proofs for why this framework can produce anomaly detection success independent of model architecture and task. They implement this method at the MNIST scale and test it against a variety of attacks, showing that is achieves impressive success.

**Audience:**

No

**Broader Impact Concerns:**

No concerns, just some notes

I appreciate the discussion of broader impact. But I think that the conclusion sounds just a little bit hand-wavy. To me, it seems to equivocate this particular anomaly detection/theory work with 'understanding' the neural network in general and appeals to some of the claims from papers in the mechanistic interpretability literature about how it's a potential solution for providing a solid foundation for research and engineering.

Overall thoughts on the paper:
I am unfortunately not fluent in relevant theory, and I have not made an attempt to assess the details behind their theorems. However, if the theory is indeed correct, I believe that this is quality work and that it would not be harmful or a mistake in any sense to accept this work. However, I ultimately am skeptical of the potential for practical impact of the work. I think that a majority of papers that focus on theory, lack experimental baselines, and experiment at the MNIST scale ultimately end up having little practical impact. My main recommendations to the authors (above) are to add baselines and scale up experiments.

**Claims And Evidence:**

Yes

**Requested Changes:**

Overall, I think that two key things would make this paper stronger.

C1: Germane mechanistic anomaly detection baselines in the experiments.

C2: Experiments to scale their method up substantially, ideally at least to the ImageNet scale or greater.

**Strengths And Weaknesses:**

S1: I commend the authors for doing theory and empirical work that is architecture and task agnostic! I think this makes the paper much stronger.

S2: I think that the depth of the experiments is very useful, and I appreciate the inclusion of attacks that seemed to beat their anomaly detection. I also have some thoughts on breadth of experimentation (see below).

S3: I have almost no complaints about the writing or presentation. I think it was easy to read through and navigate. My one recommendation would be to think about if some intuitive figure 1 could be added.

W1: From the abstract, it seems like this paper is an anomaly detection paper, but it doesn't contain the word "anomaly" and seems to not engage with literature in anomaly detection based on the related work section.

W2: The focus on MNIST and FashinMNIST seems to be a major limitation. Why not larger systems performing valuable tasks for which anomaly detection is safety/performance critical? I know it's cynical, but I am sympathetic to the perspective that anything that doesn't scale isn't likely to be practically useful.

W3: No baselines are compared against. Is there a reason I should suspect that the method would beat simple nearest neighbor, KDE, or gaussian DE anomaly detectors?

Q1: The pixel attacks are interesting! Thanks for including them. What is your interpretation of their success in light of the theorems in the paper? This is more of a question than a weakness.

Q2: I lack the theoretical fluency to engage fully with the math. I apologize. But could the authors comment on the extent to which the tightness and practical value of their theorems behind contributions 2-4 will fare with model scale?

---

> ### Author Response · Authors · 2024-10-26
> **Response to Reviewer Vj72**
>
> We thank the reviewer for the useful comments and interesting questions. We will give answers to each recommendation and question.
>
> **S3**: Thank you for the suggestion. We could add a diagram with the functions (e.g. $\Psi(W,f)$, $\pi$, $\mathrm{M}(W,f)$, etc.) between the different “spaces” that are at play (e.g. $\mathbb{R}^d, \mathbb{R}^k, \mathrm{Mat}_\mathbb{R}(k \times d)$, etc.).
>
> **W1**: For this paper, we wanted to prioritize adversarial examples more specifically. Although, we agree that it would be a good idea to mention that we use a kind of anomaly detection technique to detect these attacks (as opposed to adversarial training, for instance). Therefore, we will, like you suggested, engage a bit with the literature, at least to compare against a baseline (see W3).
>
> **W2**: At first, we wanted to introduce the approach and hopefully motivate further research. However, it became clear with the comments of the reviewers, including yours, that we need to include a wider variety of experiments to better support our theoretical results and our claims. We are currently working on it. We have run experiments on MNIST-1D [1] with CNNs, and we will run experiments on MNIST and CIFAR-10 with CNNs that we will add to the paper. The current implementation doesn’t allow us to scale the experiments substantially. Based on another reviewer’s suggestion, we want to add a section describing how to compute the matrices and therefore explain how one could improve the current implementation.
>
> **W3**: The reasoning behind this was that we didn’t want to compare against other anomaly detection methods (or any other method) used in adversarial attack detection since we didn’t want this paper to be mainly an anomaly / adversarial attack detection paper. However, we now agree that it would be a good idea to give a baseline, while still focusing on the broader idea, which is to introduce how one might use the matrices produced by $\mathrm{M}(W,f)$.
>
> **Q1**: This is a good question and we do not know the answer. What we know is that the theoretical results suggest that using the matrices is a “better” idea than using the logits to study the predictions of the neural network.
>
> **Q2**: The theoretical results obtained do not assume any task nor scale of the model. Therefore, we are not aware of the potential practical value of our results in various contexts, but we have no reason to think that it will work less well. On the other hand, by abstracting away from uses and model sizes, it is impossible to clearly answer this question. We hope that more research will be done to answer this kind of question, which we do agree is important.
>
> Finally, we want to say that we do not think that having a basic technique to detect adversarial examples means that we now have a better understanding of neural networks. However, we think that having an object that is invariant under isomorphism and quite simple to analyze is a good candidate for a tool to help us understand the behavior of neural networks. Furthermore, we hope that the fact that there is enough information in these matrices to let us detect adversarial examples indicates that we could use them for other things, like mentioned in the Discussion.
>
> [1]  Sam Greydanus, Dmitry Kobak: Scaling Down Deep Learning with MNIST-1D. https://arxiv.org/abs/2011.14439v4.

---

> > ### Comment · Reviewer_Vj72 · 2024-11-14
> > **response**
> >
> > S3: That would seem nice to try. But I'd try to make it not too mathy.
> >
> > W1: Sounds nice if you add a paragraph + a few other mentions where appropriate.
> >
> > W2: This seems ok. I would wish for more scalable techniques. Is this something you are working on in future work?
> >
> > W3: Including an AD baseline would seem good. **I think that this should be done before the paper is final.**
> >
> > Other: Thanks for responses.
> >
> > ---
> >
> > Overall, thanks to the authors. I think that this paper could be useful. I would lean toward accepting if the other reviewers concur. Consider my review to be a weak accept with a 3/5 confidence.

---

> ### Author Response · Authors · 2024-11-17
> **response to reviewer Vj72**
>
> We thank the reviewer for the response.
>
> W2: We plan to work on CNNs in future work, although it is not clear to us at the moment how to scale this for large CNNs, particularly with many channels, that is the main reason why we do not present experiments on large CNNs.
>
> W3: Yes, we will add it before the paper is final.

---

> > ### Comment · Reviewer_Vj72 · 2025-01-03
> > **Weak accept**
> >
> > I support acceptance but don't have the highest confidence. For the reasons discussed above, I think this should be accepted as long as other reviewers do not have large concerns. Thanks to the authors.

---

### Review · Reviewer_BFn5 · 2024-10-18

**Summary Of Contributions:**

The authors reformulate neural networks using quiver theory and reduce the computational complexity of the forward pass into a single matrix, enabling geometric interpretations. Furthermore, they investigate hidden layer representations through quiver representations and attempt to use those representations to detect adversarial examples.

This is the first time I have read about quiver theory and its applications to neural networks, so I really appreciated the inclusion of Section 3. That said, despite this section, I had to frequently refer to the works of [1,2] to get a proper grasp of what was going on.

[1] Armenta and Jodoin. The representation theory of neural networks.

[2] Armenta et al. Double framed moduli spaces of quiver representations.

**Audience:**

Yes

**Broader Impact Concerns:**

There are no ethical implications of this work.

**Claims And Evidence:**

No

**Requested Changes:**

**Requested changes:**

*Must haves:*

- Clarify the connection of theoretical parts to experiments.
- Use known and commonly used models for experiments.
- Use (a portion of) ImageNet dataset for experiments (you don't need to train those models, just employ pretrained models from torchvision or timm).
- Follow guidelines on evaluating adversarial attacks and defenses.
- Conduct fewer higher quality experiments instead of a large number of low quality experiments.
- Identify the limitations of the theoretical evidence in practice (e.g., why the quiver theory fails in certain scenarios).
- Modify discussion section to discuss experiments and their relevance to the quiver theory, not potential future works.
- Avoid making claims without evidence -- which is one of the primary evaluation criteria for TMLR.

*Nice to haves (but not necessary):*

- Investigate self-supervised models and identify their difference (if any) to supervised trained models.
- Investigate the relation of quiver theory and group-based accuracy of models
- Present comparative experiments on the same architecture but different training methods (e.g., ViTs trained with DINO, MAE, and supervised learning) and investigate the relation of quiver theory to training.

**Strengths And Weaknesses:**

I must say that before reading the entire manuscript, I expected the authors to move the work in the direction of analyzing representation similarities and discrepancies for semantically similar and dissimilar images, which would have made a better case for employing quiver theory to understand neural networks. To my surprise, the authors tackle the problem of adversarial examples and present experimental results based on an adversarial defense which introduced several issues and perhaps diminished the contributions of the work.

**Theoretical contributions:**

The authors claim to introduce a novel mathematical framework for analyzing neural networks using tools from quiver representation theory; however, this is (mostly) not true. The majority of the theoretical results in the work are already published in [1,2]. I suggest that the authors exercise caution with their choice of wording, as they build on or extend the work of [1,2], and do not propose the entire framework, let alone introduce it for the first time.

Continuing on the theoretical side, most of the theoretical parts introduced in the work are straightforward extensions of the findings from [1,2]. Additionally, the outcome of Theorem 4.4, described in the following sentence, is vague, with hand-wavy definitions such as "the accuracy both on train and test data is high," "the loss both on train and test is sufficiently low," and "if a neural network is well-trained.".

**Bridging theoretical observations and adversarial examples:**

The work does a poor job of relating the theoretical observations to adversarial examples. From Section 3 (page 3) to Section 5 (page 7), where the theoretical foundations are laid, adversarial examples are not mentioned once, except at the very end, where the authors state that they will use the theoretical foundations to detect adversarial examples. Then, in a sudden shift in tone, the authors describe adversarial attacks and networks in Section 5.1. This section should be improved for clarity, and the relation between the theorems and adversarial examples/defenses should be described in Section 4.

**Applicability of theoretical contributions:**

The neural networks considered in this study are only MLPs, which on its own is not a problem for investigating representations through quiver theory. However, the wording used by the authors in the abstract and the rest of the manuscript gives the impression that this framework can be applied to any neural network, which may not be so straightforward. Furthermore, the authors frequently reference LLMs throughout the manuscript, yet they do not present any investigation on those architectures. It might be a good idea to move those discussions to a section on future work. Another interesting observation is the lack of results on CNNs, which are still the dominant architecture in computer vision. The authors do not present a single result or discuss the relation of their observations to such models, except in Section 3, where they imply that their theoretical observations might be harder to apply to DenseNets. It is also strange to see experiments on MNIST and FashionMNIST but without CNNs such as a simple LeNet.

**Experimental results:**

At a first glance, it seems like there are a lot of experiments in this work, but it is not true. The authors simply used all attacks on all models and evaluated their defense on those adversarial examples. There is no insight as to why the quiver representation works or fails to work in certain cases. Furthermore, the experimental setup is not appropriate, as it basically falls into every pitfall known to the community that investigate adversarial attacks/defenses.

**Detecting adversarial examples:**

Experimental design to detect adversarial examples is not appropriate for a study conducted nearly a decade after the initial investigations into adversarial examples. Major issues in this section include:

*- Datasets:*

MNIST and FashionMNIST are known to lead to potentially misleading conclusions when working with adversarial examples, both due to their simplicity and the homogeneity of their datasets [3,4]. The authors should select datasets that are relevant to the TMLR community such as ImageNet.

*- Models:*

The evaluated models are arbitrary. The authors should select models that are commonly used and well-known in the field (for example: ResNets and Vision Transformers).

*- Selection of attacks and attack parameters:*

Details on the attacks and the capability of the attacker is not clearly defined. Is the tackled scenario a black- or white-box setting? If it's white-box, do the authors assume the attacker only has access to the model or the subsequent quiver representations as well? Many (almost all) details are missing from the work.

*- A final remark on the aforementioned criticism:*

I understand that the primary focus of this work is not to propose a novel adversarial defense. That said, the authors chose this direction to present experimental results. In that case, they should follow guidelines established in the field for this work to be relevant to readers of TMLR. More details on those can be found in [3,4,5], as well as in state-of-the-art defenses in this benchmark [6].

**On Discussion:**

In Section 7 (Discussion), the authors discuss topics that are completely irrelevant to the experimental setting/results such as (1) OOD detection, (2) the applicability of observations to LLMs, and (3) the generalization of neural networks to unseen (OOD) samples. Each of these topics is a major research area in its own right, and this discussion is mostly unsubstantiated. In particular, the discussion regarding LLMs (in Section 7 as well as other sections) has no place in this manuscript when all that is analyzed are simple MLPs. It gives the impression that the authors are attempting to capitalize on the most recent topic (LLMs) without putting effort into investigating those models, most likely due to their size and the difficulty of experimentation.

This section should focus on the observations made in the experimental results and their relevance. Why the proposed defense failed for some images? Is it because of the models, images or something else? Are there some properties the quiver representation fails to capture?

**Final remarks:**

Especially towards the end, this manuscript suffers from trying to tackle too many angles I described above such as adversarial examples, feature similarity, relevance to LLMs and OOD samples, generalization. Furthermore, experimental setting fails to prove the point for the theoretical observations and leaves much to be desired.

[3] Carlini et al. On Evaluating Adversarial Robustness

[4] Carlini et al. Adversarial Examples Are Not Easily Detected: Bypassing Ten Detection Methods

[5] Carlini et al. Towards Evaluating the Robustness of Neural Networks

[6] Croce et al. RobustBench: a standardized adversarial robustness benchmark -- https://robustbench.github.io/

---

> ### Author Response · Authors · 2024-10-28
> **Response to Reviewer BFn5 (I)**
>
> First, we want to thank the reviewer for the many useful suggestions and comments. As a general comment, we want to stress, as you acknowledge, that the primary goal of this paper was not adversarial detection but to motivate the use of the matrices to study the predictions of neural networks. This is the main reason why our experiments were lacking on many fronts. We understand your concerns and we want to address them by improving our experiment setup.
>
> **Theoretical contributions:** The framework proposed is not the use of representation theory of quivers in the study of neural networks, which was indeed introduced before this paper. The framework proposed is the use of the matrices induced by the quiver representations and then the use of statistical and geometric arguments to study the predictions of neural networks which is, as far as we know, a novel approach. We can however change the wording to make sure this is clear. Also note that we do give credit to the works of [1,2] on every single one of their results and we do not claim ownership of any of their results.
>
> About the outcome of Theorem 4.4, we could clarify by making specific claims to our experiments, for example, experiment 12 has 91% train accuracy which means that 91% of the training data $x$ is mapped by $\mathrm{M}(W,f)$ to their corresponding partition $\mathcal{M}_j$ in matrix space. This is what we meant by our wording on “well trained” and “high accuracy” or “sufficiently low loss”.
>
> **Bridging theoretical observations and adversarial examples:** The theory suggests that new samples can be trusted if their matrices are closer to the matrices of the training data and it is this idea that we apply to detect adversarial examples using these matrices. We can add a clarification of this as the reviewer suggests. We also want to emphasize that due to the theoretical results, we can be sure that the training data is mapped by $\mathrm{M}(W,f)$ to their corresponding class partition as mentioned above, which allowed us to come up with the class dependent hyper-ellipsoids that we use to detect adversarial examples. It is worth noting that we do nothing to the training of the neural network nor train other networks for detection, as done by adversarial training and class conditional GANs [a], for example. Based on the recommendation of another reviewer, we want to make the last point more clear by engaging with the literature on anomaly detection.
>
> **Applicability of theoretical contributions:** The theory we provide applies to any architecture, since we do not make any assumption on the architecture in any of our results. However, we do agree with your comment that it is not straightforward how to use this framework, let alone compute the matrices. We will add a section on the computation of the matrices, where we will discuss the subtleties with different architectures and also the cost of the computation.
>
> Like said earlier, our main focus was not on the empirical results and this is why our experiments were small and deficient in different ways. However, we understand your comments and ultimately are sympathetic with the idea that a more diverse set of experiments would be much more interesting. We want to add, like you recommended, experiments with different architectures and other datasets. We are running experiments with small 2D CNNs on CIFAR-10, and also 1D CNNs on MNIST-1D [b]. We will add these results to the paper. Our current implementation does not allow us to compute the matrices for 2D convolutions with more than 32 channels because of memory issues. An efficient implementation is not our goal for this paper, but we believe that a much more efficient algorithm could be implemented. We see room for improvement although not trivially. For now, this makes doing the experiments on well known architectures unreachable and this is why our models may seem arbitrary.
>
> **Experimental results:** We are thankful for the many suggestions and resources you gave us about improving this section. Like mentioned in the previous section, we will add experiments with different architectures and datasets. Also, we will include other types of attacks and include more details on the capability of the attacker. As we said before and you acknowledged, we do not want this paper to be strictly about adversarial detection, but we understand that a wider and more appropriate set of attacks and including more details on the experiments, along with more datasets and architectures, is important. We also think that this can be done while still focusing on our main goal, which is to introduce how one might study neural network predictions by studying the matrices produced by $\mathrm{M}(W,f)$.

---

> > ### Author Response · Authors · 2024-10-28
> > **Response to Reviewer BFn5 (II)**
> >
> > **On Discussion:** We propose to change the section title to "Future Work". Based on the main objective of the paper, we think this section is important to encourage further study based on these matrices. Also, concerning OOD detection, we can include results on small experiments, to at least give an intuition behind our proposed algorithm, where we can detect images from, for example, MNIST fed to a neural network trained on FashionMNIST.
> >
> > About LLMs, we first mention them in the Previous Work section to emphasize how important adversarial attack detection is. Then, we mention them in Experimental Results to give an example on when it might be a good idea to reject more potentially “clean” sample. This can be removed if it gives the sense that our results on adversarial detection could directly be applied to LLMs, which is certainly not what we were trying to do. Finally, we mention LLMs in the Discussion section. There, we state that we propose to analyze the MLPs that are inside these transformers. We think that this is supported by our theoretical and experimental results.
> >
> > **Requested changes:** Concerning the claims we make in the paper, we can change the first sentence of the abstract to read “We introduce a novel mathematical framework for analyzing neural network *predictions*…” which will avoid the confusion that we introduce the whole framework of quiver representations for neural networks. As well as in the first sentence of the second paragraph of the introduction to “analyzing neural network *predictions*…”. Finally on the last sentence of the previous works section to “for studying and understanding neural network *predictions*”. We want to stress that the rest of our claims are backed up by theorems and experiments. We have addressed the rest of the requested changes above.
> >
> > [a] Hang Wang, David J. Miller, George Kesidis: Anomaly detection of adversarial examples using class-conditional generative adversarial networks, Computers & Security, Volume 124, 2023, 102956,ISSN 0167-4048, https://doi.org/10.1016/j.cose.2022.102956.
> >
> > [b]  Sam Greydanus, Dmitry Kobak: Scaling Down Deep Learning with MNIST-1D. https://arxiv.org/abs/2011.14439v4.

---

> ### Comment · Reviewer_BFn5 · 2024-11-15
> **About proposed revisions**
>
> I would like to thank the authors for their detailed response to my comments. Most of the proposed changes sound appropriate to me, except for the experimental results.
>
> If the proposed technique is not applicable to any model with a convolutional channel larger than 32, it is effectively inapplicable to any CNN used today, including AlexNet (which was proposed more than a decade ago). Please correct me if I'm wrong, but this would also make it extremely unrealistic to consider using it for any transformer-based model, as those models have even more parameters than CNNs. In light of this, I suggest that the authors clearly mention this limitation.
>
> My main concern with this work is that its experimental findings are not relevant to the vast majority of TMLR readers, as most published research (on vision) focuses on state-of-the-art (SOTA) models. There have been numerous theoretical findings that worked for simple models but failed for larger ones either due to technical limitations or theoretical assumptions and, based on the authors comments, this work will be just another one. As for the models, the authors should at least use some of the well known models such as LeNet, at the very least, to present any compelling evidence. Additionally, the experimental setup for the adversarial attack study is not appropriate. While I understand that this is not the main focus of the paper, it is still a part of it. Authors should pay attention to best-practices (beyond the selection of models and attacks) in the field and revise their experiments accordingly.

---

> > ### Author Response · Authors · 2024-11-17
> > **response to reviewer BFn5**
> >
> > Thank you for the response. Before answering, we want to say that the current limitation you mentioned will indeed be addressed in detail in the new section on the algorithm's complexity and the new Discussion section.
> >
> > About the current problem with CNNs, we want to clearly state that this is *only* due to the way we implemented the algorithm to compute $\mathrm{M}(W,f)(x)$ and not because of the number of parameters. The problem (and where the algorithm could be substantially improved) will be apparent in the new section on the matrix computation. Our implementation converts a convolutional layer to a big sparse almost-circulant matrix (it is not circulant because of the padding) and performs a matrix product on these huge matrices for which the sparse methods do not solve the memory problem. This is why our current implementation does not scale for 2D convolutional layers only.  The problem is not the number of parameters, for instance, in the case of MLPs, the algorithm used is much better and we had no problem computing the matrices for a neural net of about 2 billion parameters (see Table 1 in the paper). For the case of transformers that do not include 2D convolutional layers, the matrices can indeed be computed and it is the type of research we want to motivate and work on in the future. This being said, we will add experiments on 2D convolutional networks on CIFAR-10 with 10 channels.
> >
> > We remark that we do not claim at any moment that we develop an adversarial example detection algorithm that compares to SOTA. We claim that by using the matrices we can detect several adversarial examples coming from different attacks (we will add more different attacks in the new version) by a simple statistical and geometrical analysis in matrix space. The evidence we provide does back up this claim.

---

### Review · Reviewer_Kv2C · 2024-10-21

**Summary Of Contributions:**

This paper makes several contributions. Firstly, it provides a new technical framework to study the robustness properties of neural networks using quiver representation theory. The authors show that matrices produced by quiver representations of neural networks are invariant under network isomorphisms and that matrix distances can provide lower bounds for the distance between network logits, regardless of architecture, training, or task. Secondly, the authors propose a novel adversarial example detection method based on these matrix properties. Using the MNIST and FashionMNIST datasets, several MLP architectures, and a set of adversarial attacks, they demonstrate the effectiveness of this novel detection method in flagging adversarial inputs.

**Audience:**

Yes

**Claims And Evidence:**

No

**Requested Changes:**

While the overall idea is novel and interesting, it is unclear how usable and effective the proposed detector really is. Therefore, I request the following changes:

## Better highlight how usable (or, practical) the proposed method is.
1. Add results using datasets that are better representative of real-world images (like cifar10 and imagenet), and using models that are actually employed for image data (like CNNs and transformers).

2. Please add some results regarding how much computational overhead the detection method introduces. This will not only highlight the usability of the detector, but also set up a direction for potential improvement via future work.

## Adequately establish how effective (or, robust) the proposed method truly is.
1. Perform evaluation using a more diverse set of attacks. I recommend removing some of the PGD-derivative attacks and adding meaningfully different ones like gradient-free ones. For example, using all the attacks in the AutoAttack ensemble should be sufficient.

2. For papers proposing defenses whose robustness is empirical in nature, it is critical that adaptive attacks are giving proper attention during evaluation. Off the top of my head, the adversary can simultaneously target the classifier and the detector's objective. Please address such an adaptive attack. If it can't be done, explain why. If it can be done, please present the corresponding results.

3. Please compare against at least one state-of-the-art detector. The purpose here is NOT to show whether the proposed method beats the state-of-the-art or not, but to put into perspective how it compares with what's already out there.

**Strengths And Weaknesses:**

## Strengths

1. Authors apply a novel mathematical tool towards detection of adversarial samples, namely quiver representation theory.
2. The theoretical results are backed up with proofs.
3. The empirical results show great potential of the applicability of quiver representation theory towards improving adversarial robustness.

## Weaknesses

The theoretical parts are dense and hard to follow, especially for someone not familar with quiver representation theory.

The experiment section is lacking on several fronts:

1. Results are presented on two very simple datasets only. While using datasets like MNIST/FMNIST is not discouraged, only having them in the experiment sections is not enough. Having at least one dataset that is more representative of real world images like cifar10/imagenet is essential to establish the true usability of the proposed method.

2. Authors present results using only MLPs, which are not well-suited for image-based tasks. Since the paper exculsively focuses on image classifiers, it is critical to evaluate how effective the proposed method is when used with popular vision models like CNNs and, more recently, transformers.

3. The detector is evaluated against a set of redundant attacks which don't effectively capture the threat model assumed by evasion attacks. Majority of chosen attacks are based on PGD, a gradient-based iterative attack. Furthermore, it doesn't include any gradient-free attacks like boundary attack [a], square attack [b], or transfer-based attacks [c]. As such, it is unclear how truly effective the detector is against evasion attacks. At the very least, having all attacks from the AutoAttack [d] ensemble would add sufficient credibility to the results.

4. The authors do not consider an adversary that is aware of the detection method in place. One straightforward strategy such an adversary can employ is to attack the classifier's and the detector's objective collectively. In the absence of evaluation against such adaptive adversaries, or a discussion regarding why such adaptive attacks can't be done, it is unclear how effective the detector really is against evasion attacks. In addition to the adaptive attack I have described here, motivation for other adaptive attacks can be obtained from prior works [e] that specifically design attacks to bypass detection-based defenses.

5. The authors do not make any comparisons with other detection-based defenses.

## References
[a] Decision-based adversarial attacks: Reliable attacks against black-box machine learning models. ICLR 2018.

[b] Square Attack: A Query-Efficient Black-Box Adversarial Attack via Random Search. ECCV 2020.

[c] The space of transferable adversarial examples.

[d] Reliable evaluation of adversarial robustness with an ensemble of diverse parameter-free attack. ICML 2020.

[e] Adversarial examples are not easily detected: Bypassing ten detection methods. ACM workshop on AI and Security 2017.

---

> ### Author Response · Authors · 2024-10-28
> **Response to Reviewer Kv2C**
>
> We thank the reviewer for the comments, the useful suggestions, and the many references provided to help this paper.
>
> **Weaknesses:** Based on your comment and one made by another reviewer, we propose to add a diagram with the functions (e.g. $\Psi(W,f), \pi, \mathrm{M}(W,f)$, etc.) between the different “spaces” that are at play (e.g. $\mathbb{R}^d, \mathbb{R}^k, \mathrm{Mat}_\mathbb{R}(k \times d)$, etc.). We believe this could help the readers to follow more easily the theoretical part of this paper.
>
> **1,2,3:** We first want to stress that our main objective for this paper was to introduce a novel framework, namely the use of the matrices to study neural network predictions. We ended up studying adversarial detection because we thought it is interesting that very small changes in the input could be detected in a predictable way in the matrices. With that being said, we ultimately agree with your comments and recommendations and we are currently working on adding experiments on a wider variety of architectures, datasets, and attacks.
> We are running experiments on small 2D convolutional neural networks trained on CIFAR-10 and MNIST and 1D CNNs on MNIST-1D [1] with a wider set of attacks, including the attacks from AutoAttack. The main reason to focus on small architectures is that our current implementation does not allow us to compute the matrices for 2D CNNs with more than 32 channels because of memory issues. An efficient implementation is not our goal for this paper, but we believe that a much more efficient algorithm could be implemented and we see room for improvement (see Requested Changes).
> Finally, we do not claim the detection method has any usability, rather, that arguments using the matrices can be powerful enough to detect adversarial examples, and then encourage further research using these matrices.
>
> **4:** Indeed, we do not consider this scenario. But we also do not claim that our detection method can beat any type of attacker. For instance, we show attacks that fool our detection method. On the other hand, we claim that the matrices give more information than the output of the neural network and that simple statistics in matrix space allows us to detect a wide variety of attacks.
>
> **5:** Like mentioned to another reviewer, the reasoning behind this was that we didn’t want to compare against other detection methods used in adversarial attack detection since we didn’t want this paper to be mainly an adversarial attack detection paper. However, we now agree that it would be a good idea to give a baseline, while still focusing on the broader idea, which is to introduce how one might use the matrices produced by $\mathrm{M}(W,f)$ to study neural network predictions.
>
> **Requested Changes:**
> The only suggestion that was not addressed is the one concerning the computational cost of computing the matrices. We are thankful for this suggestion and we think this is a very good idea. We will include a section on the computation of the matrices. We will discuss the computation with different architectures and also give details on the computational cost and potential improvement.
>
> [1]  Sam Greydanus, Dmitry Kobak: Scaling Down Deep Learning with MNIST-1D. https://arxiv.org/abs/2011.14439v4.

---

> ### Comment · Reviewer_Kv2C · 2024-11-17
> **Post-rebuttal response**
>
> Adding a diagram to explain the technical prerequisites will definitely help digest the information better. Thank you for adding it.
>
> 1,2,3: I understand, thanks for clarifying. Since the experiment section is predominantly on detecting adversarial attacks, I believe it should be judged on the same standard as other adversarial ML papers in terms of technical soundness. In the case of this paper (Adv ML is not the main focus), relaxation in expectations may only apply when judging whether proposed method improves upon SOTA or not. Adding results that you have mentioned will definitely help improve the soundness of the experiment section and I am looking forward to the results in the next iteration of the paper.
>
> 4: In that case, what is the value of a method that can detect adversarial examples but only in very specific situations? In my opinion, it is not as interesting if a method (not designed for attack detection)  can detect attack in very specific (non-adaptive) settings. Irrespective, it is important to highlight this limitation in the paper.
>
> 5: If you don't want the paper to primarily focus on adversarial attacks, please diversify the experiment section a bit. For example, free up space by moving Table 3 in appendix and only present a subset of attack detection results in the main paper. Use the remaining space to show other application of the proposed method (eg, OOD detection and generalization).

---

> > ### Author Response · Authors · 2024-11-19
> > **response to rebuttal reviewer Kv2C**
> >
> > Thank you for the response.
> >
> > 4: We claim the value of the method is to show that the matrix associated with the forward pass (i.e. $\mathrm{M}(W,f)(x)$) has enough information to detect changes in the input space (see 5 for further details). We will indeed highlight the limitation you mentioned, namely that we do not consider an adversary that is aware of our detection method.
> >
> > 5: Based on another reviewer's comment, we had already planned on adding results on OOD detection. We agree with your comment that this will better emphasize that our goal is not specifically to detect adversarial attacks. Moreover, we think that this will give further evidence that these matrices can be an interesting tool of study.

---

### Decision · Action_Editor_t5mo · 2024-11-23

**Recommendation:** Reject

**Comment:**

This is a hard decision. The paper shows potential and seems to bring new ideas related to robustness. However, reviewers have raised concerns over the settings and whether the theoretical results are verified by the experiments. As such, I would recommend to the authors to improve their paper and complete the extensive revision required before resubmitting to TMLR (if this is of interest of course). In addition, the authors are recommended to follow the standard setups for those experiments, since validating the theoretical results using a standard setting is important.

**Audience:**

Understanding MLP layers, robustness are definitely at the core of machine learning and as such this paper could be relevant to TMLR.

**Claims And Evidence:**

The paper introduces a new framework using quiver representation theory to study neural network robustness. The authors propose an adversarial example detection method based on these matrix properties, demonstrating its effectiveness on various toy datasets across various MLP architectures and adversarial attacks.

The reviewers have raised concerns over the empirical evidence and observations and whether those settings are following the standard principles in machine learning.

**Resubmission Of Major Revision:**

The authors may consider submitting a major revision at a later time.